



# Long-term ensemble forecast of snowmelt inflow into the Cheboksary reservoir under the differently constructed weather scenarios

Alexander Gelfan[1], Vsevolod Moreydo[1], Yury Motovilov[1]

[1]Institute of Water Problems of the Russian Academy of Sciences, 3 Gubkina str., 119333 Moscow, Russia

*Correspondence to*: Alexander Gelfan (hydrowpi@iwp.ru)

**Abstract.**

Development and verification of the long-term ensemble forecast of snowmelt water inflow into the Cheboksary reservoir - one of the eleven major river reservoirs of the Volga-Kama Reservoir Cascade, are described. The used forecasting procedure is based on a combination of semi-distributed hydrological model ECOMAG that allows for ensemble calculation of inflow hydrographs with two types of weather ensembles for the lead-time period: (1) observed weather data constructed on the basis of the ESP methodology and (2) synthetic weather data simulated by a weather generator. Ensemble forecasts represented both in deterministic and probability forms were verified by producing hindcasts of water inflow into the Cheboksary reservoir for 35 seasons in 1982 – 2016 from the 1st of April for 3 months ahead. Four inflow characteristics were forecasted: volume and maximum discharge of the inflow, as well as amount of days with inflow above two specified thresholds. Results of operational forecast of the recent freshet of 2017 are additionally presented. We found that statistical distributions of the forecasted ensembles calculated under both types of weather scenarios are very close to each other. However, confidence intervals of the forecast statistics is much larger for the ESP-based ensemble. Sensitivity of the forecast performance to changes in the forecast horizon and issue date is analyzed and so-called "forecastability maps" are presented as useful tool for such analysis.

## 1. Introduction

Spring freshet is a hydrological phenomenon, which scale is highly dependent on the amount of water accumulated in surface and subsurface storages of the river basin during several months prior to the snowmelt. This dependence serves as a physical basis to predictability of spring runoff (Li et al., 2009) and creates an opportunity for its forecasting with the lead-time subject to water storage release conditions after the forecast issue date. As stated by Lettenmaier and Waddle (1978, p.1) "snowmelt runoff is one of the few natural phenomena for which relatively accurate long-term forecast can be made". Implementation of this opportunity is crucial for the water reservoirs of the Volga-Kama Reservoir Cascade (VKRC) – one of the world's largest multi-purpose water management systems. The VKRC is situated within the largest European Volga River basin (area of 1 350 000 km$^2$) and consists of 11 reservoirs that hold from 1 to 58 km$^3$ of water and conduct seasonal



and multiyear flow regulation. The VKRC was designed to redistribute the highly uneven runoff of the Volga River, with 2/3 of the annual runoff volume occurring during two to three months of the spring freshet. This task, aimed at optimizing the water supply for power production and waterways navigation as well as flood protection, is even more complex due to the requirement of annual spring water release to Lower Volga in order to allow for sturgeon spawning. Such several-weeks-long regulated release with predefined amount of water during the spring freshet is a unique task for a water management system (Avakyan, 1998). Hence, extension of the lead-time of reservoir inflow forecast during the spring freshet is crucial for the decision makers.

The first long-term forecasts of the spring water inflow into the VKRC reservoirs are dated back to 1930s – 1940s, when the first reservoirs were filled to capacity. By 1970s the main approaches and specific forecasting methods were developed, which underlie the contemporary operational forecasting. These essentially deterministic approaches can be divided into three groups: (1) direct water balance calculation methods, (2) index methods, and (3) physical-statistical methods. The first group of methods is based on direct measurements of the water balance components and has very limited applicability (Borsch, Simonov, 2016). Gelfan and Motovilov (2009) give a detailed review of the index and physical-statistical methods. In general, these methods relate the spring runoff volume to a number of basin characteristics prior to the snowmelt offset (such as snow water equivalent, soil freezing depth and soil moisture content) and precipitation amount for the forecast lead-time. These characteristics are assigned by observations, yet the precipitation amount for the forecast lead-time is set as climatic mean.

The increase of demand for economic efficiency and safety of water management systems specifies the necessity to improve the existing methods of long-term forecast of water inflow into reservoirs, increase its accuracy, lead-time and informational content. At the same time, the improvement capacity of the regression-based forecast dependencies is limited - mostly due to corruption of homogeneity in the streamflow observation data and runoff predictors that occurred since the forecast methodology developed. Such corruption can be caused either by reduction of the observational network (estimated at 30% in (Borsch, Simonov, 2016)), or improvement of measurement techniques (shifting towards remote sensing, in particular), or water regime alterations of the Volga river due to global and regional climate change (Dzhamalov et al. 2014).

The capacity to improve the existing methods of long-term forecasting of water inflow into VKRC is in shifting from the exclusively regression-based forecasts towards hydrological model-based forecasts and from deterministic methodology to the ensemble one (Pappenberger et al., 2016). Utilizing the process-oriented distributed hydrological models results in strengthening of physical adequacy of the forecast and, potentially, in improving forecast accuracy and lead-time, as well as the forecast informational content. For instance, it may be possible to predict inflow time-series rather than just the runoff volume, to assimilate spatially distributed data (e.g. remote sensing data, seasonal weather forecast) etc. (see Wood and Lettenmaier, 2008; Nagler et al., 2008; Li et al. 2013). Utilizing the ensemble methodology allows forecaster to provide user not only the most likely value of the forecasted variable, but also yield a range of possible values, account for the forecast uncertainty and  dispel, thereby, the "illusion of certainty in a user's mind" (Krzysztofowicz, 2001, p.3). Recent studies illustrating ability of the ensemble, hydrological model-based forecasts to increase the decision making efficiency of



reservoir management and operations, including snow dominated basins, are presented, for instance, by Graham and Georgakakos (2010), Georgakakos et al. (2012), Zhao and Zhao (2014), Brown et al., (2015), Anghileri et al. (2016).

In the early 2000s, the Integrated Modelling and Reservoir Management Complex (IMRMC; Motovilov et al., 2003) was developed and adopted by the Federal Water Resources Agency of the Russian Federation for so-called "scenario
calculation" of inflow into the reservoirs of the VKRC. The IMRMC integrates two main components. The first one is a forecasting system that comprises the physically-based hydrological model ECOMAG (ECOlogical Model for Applied Geophysics; Motovilov et al., 1999) in combination with the Ensemble Streamflow Prediction (ESP) procedure (Day, 1985) to construct possible weather scenarios for the lead time of the forecast. The second component of the IMRMC is the reservoir operation modelling system(Bednaruk, 2011). Both components are described in (Gelfan and Motovilov, 2009).
The ECOMAG-based forecasting component of the IMRMC has been improved during the recent years in the Water Problems Institute of the Russian Academy of the Sciences (see Gelfan et al., 2015). It has been used by the Hydrometeorological center of Roshydromet (the Russian executive authority in the field of hydrometeorological forecasts) in test mode since 2016 (Borsch and Simonov, 2016).

One of the feasible directions of the IMRMC forecasting system development relates to expansion of the weather scenarios
construction for the forecast lead-time as compared to the original ESP methodology. The latter is based on the assumption that the weather scenarios, which were previously observed during the forecast lead-time period, are representative to be set as an ensemble of future weather conditions after the forecast issue date (Day, 1985). The streamflow ensemble calculated by driving a hydrological model with the ensemble of weather scenarios is used to assess the probability distribution of the desired characteristics and issue a probabilistic forecast. The described methodology for creating a long-term forecast is well
established (see, for instance, recent review in Arnal et al. (2017)) and is currently used in operating systems in a number of countries (Pappenberger et al., 2016). However, the observed weather scenarios that are used to drive the hydrological model do not encompass all of the possible weather conditions for the forecast lead-time; it is desirable to account not only for the observed weather, but for possible weather condition that might lead to freshet events of rare occurrence. Assessing the magnitude of such an event might be crucial for decision making.
Moreover, hence the ensemble size is limited to a number of the observed weather scenarios,  the sample variance problem escalates with respect to statistics estimated from small ensemble and, as a result, an expansion of the confidence interval of the probability function quantiles estimates occurs. This problem, in turn, may generate large sample variance of the forecast verification metrics' estimates and, consequently, difficulties with the forecast interpretation by a decision maker.

The mentioned problems, both arising from a limited number of the observed weather scenarios, can be solved by
incorporating a stochastic weather generator (WG) that will allow for reproduction of a hydrological system response to a large variety of possible weather conditions for the lead-time period, to calculate large ensemble of forecasts and assess their probability properties.

Hanes et al. (1977) were probably the first who used Monte-Carlo simulated sequences of daily precipitation to drive the conceptual US Geological Survey model of runoff generation and provide ensemble long-term forecast of snowmelt freshet



runoff volume. A physically-based distributed model was used in combination with a weather generator to create a long-term probabilistic forecast of spring runoff of rivers in Central Russia in (Kuchment and Gelfan, 2007).

The purpose of this paper is to present the performance assessment of a long-term ensemble forecasting system of water inflow into the Cheboksary reservoir of the VKRC. The system uses two ways of weather ensemble construction for the forecast lead-time: observed weather data on the basis of the ESP-methodology (hereafter, ESP-based method) and synthetic weather data simulated by a weather generator (hereafter, WG-based method). The first evaluation of this system was presented in (Gelfan et al., 2015), where the ability for improving of binary forecast of maximum inflow into a reservoir by using the generated weather scenarios was shown. This paper shows verification of deterministic and probabilistic forecasts for several characteristics of inflow into the reservoir (volume and maximum discharge, amount of days with inflow above certain thresholds) and its performance under both aforementioned approaches for assigning weather scenarios for the forecast lead-time.

The paper is organized as follows. In the next two Sections, the case study basin as well as modelling tools (deterministic hydrological model and stochastic weather generator) are presented. Next, two forecasting schemes adopted to two different weather ensembles and the used verification metrics are described. The forecast verification results as well as operational forecast application to the recent freshet of 2017 are demonstrated in the fifth Section. Also, analysis of sensitivity of the forecast performance to changes in the forecast horizon and issue date is described in this section. The overall conclusions are given in the last Section.

## 2. Case study basin

The Cheboksary reservoir is located on the Volga River in central part of European Russia. It was constructed in 1982 to become the 11[th] member of Volga-Kama reservoir cascade with Nizhegorodskoe reservoir upstream and Kujbysevskoe reservoir downstream of it. Total unregulated basin area of the Cheboksary reservoir is 373 800 km$^2$ (Fig. 1). Its main tributaries - Oka, Sura and Vetluga Rivers - account for 80 to 90% of annual inflow into the reservoir.





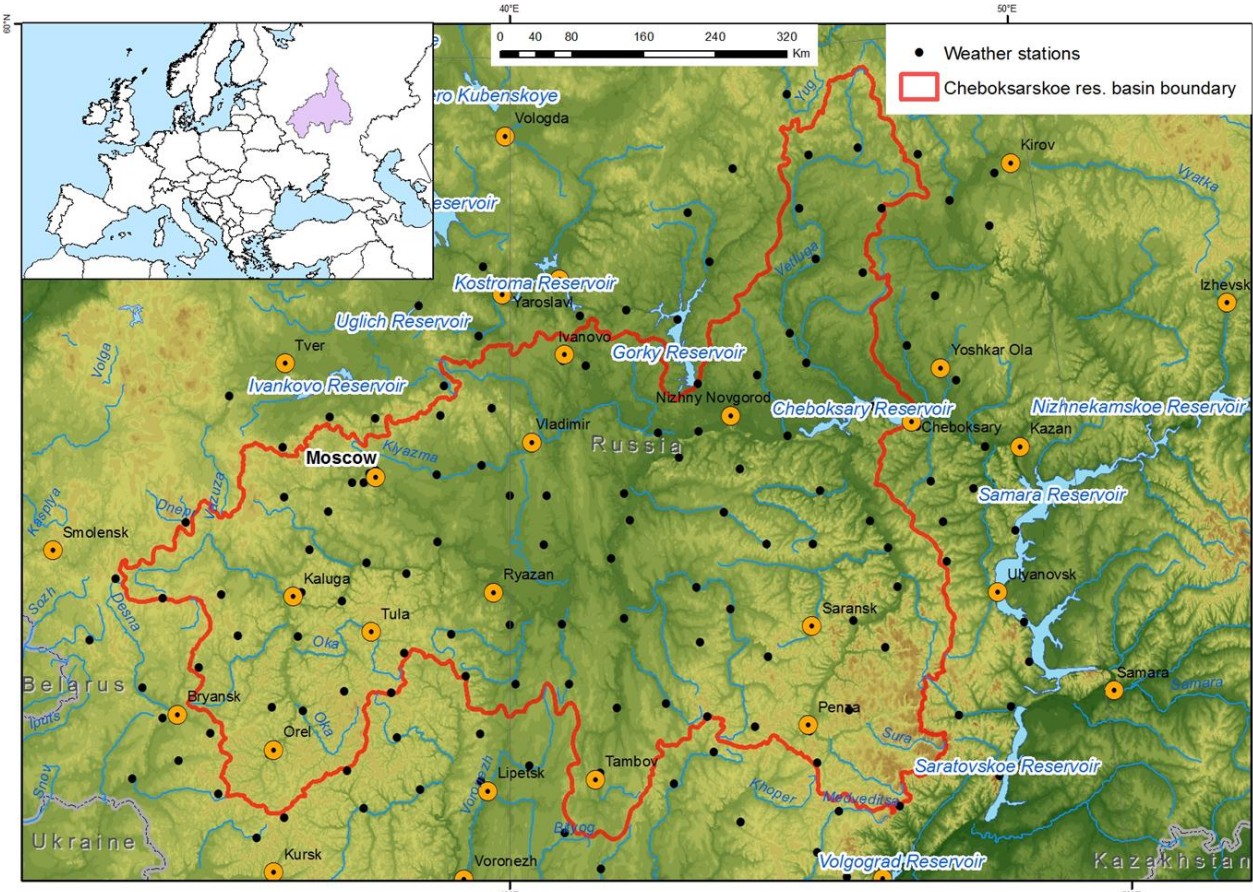

**Figure 1 – Cheboksary reservoir basin: topography, river network, weather stations**

Local climate conditions can be described as moderately continental, with cool snow-abundant winter and relatively hot summer. Mean annual temperature ranges from 1.4°C in the northern part of the basin to 4.8°C in the southern part. During wintertime air temperature may fall as low as -35 - -40°C. Annual precipitation amount ranges between 650 and 750 mm throughout the territory. Around 60% of the precipitation occurs as rain. Still, most of the winter precipitation is stored as snow cover, emerging in mid-December and lasting until mid-April. Snow water equivalent ranges from 50 mm in the South-Western part up to 100 – 120 mm in the North. Springtime snowmelt contributes to high-flow freshet – the dominating hydrological season accounting for around 65% of the total annual inflow into the reservoir (51.3 km$^3$). Freshet in average commences in mid-April and lasts until June. Mean volume of inflow for the period of reservoir operation (1982-2016) is 33.4 km$^3$, mean maximum inflow discharge is 9355 m$^3$/s.





## 3. Methods: models, forecasting procedure and verification metrics

### 3.1 Hydrological model (ECOMAG)

The ECOMAG (ECOlogical Model for Applied Geophysics; [Motovilov et al., 1999]) is a semi-distributed process-based hydrological model describing processes of snow accumulation and melt, soil freezing and thawing, water infiltration into

unfrozen and frozen soil, evapotranspiration, thermal and water regime of soil, overland, subsurface and channel flow on a daily time step (Fig. 2). The model accounts for measurable watershed characteristics such as surface elevation, slope, aspect, land cover and land use, soil and vegetation properties. The parameters are spatially distributed by partitioning the watershed into sub-basins (elementary basins) that are simulation units of the model. Parameterization of subgrid processes is described in (Motovilov, 2016). Most of the parameters are physically meaningful and can be assigned from literature or

derived through available measured characteristics of topography, soil, and land-cover. The model is driven by time series of air temperature, air humidity and precipitation intensity daily values.

The model is calibrated against measurements at different streamflow gauges, as well as available measurements of the internal basin variables (snow characteristics, soil moisture, groundwater level, etc.). The ECOMAG calibration procedure is described in detail by Gelfan et al. (2015). The model was earlier applied for hydrological simulations in many river basins

of very different size and located in different natural conditions: from small-to-middle size European basins (Gottschalk *et al.*, 2001; Gelfan et al., 2015) to the large Volga, Lena, Mackenzie and Amur Rivers with watershed areas exceeding a million km$^2$ (Motovilov 2016; Gelfan et al., 2017; Kalugin, Motovilov, 2017).

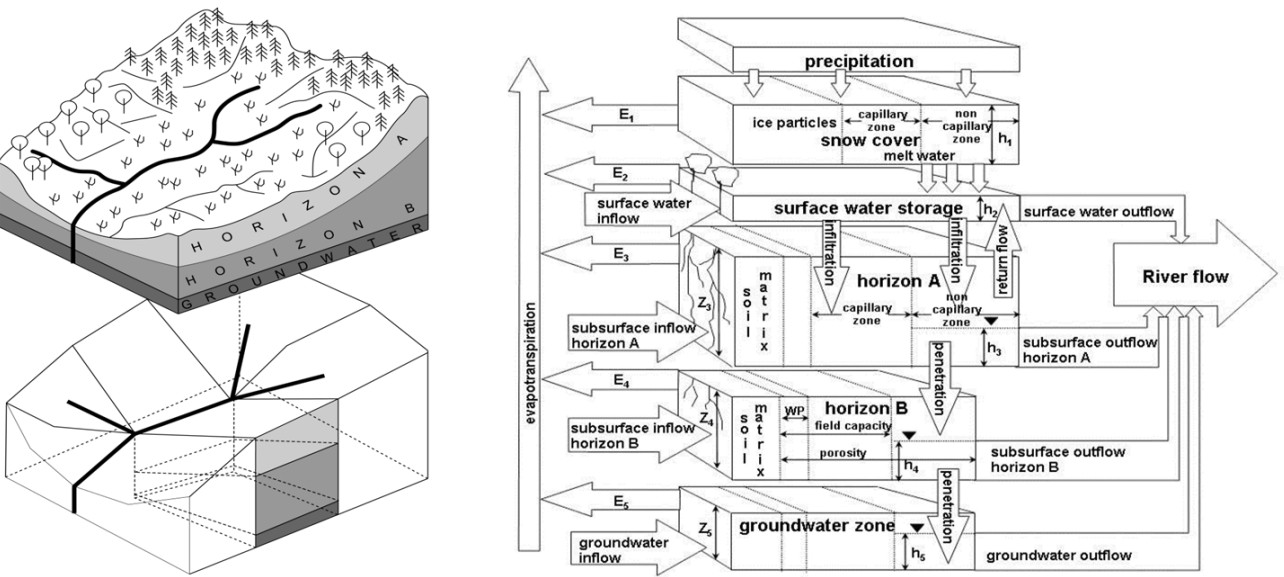

**Figure 2 – The ECOMAG basin schematization and model structure**





 In this study, 1x1 km spatial resolution DEM was used, from which stream network and elementary basins (see above) were calculated, using ECOMAG model extension for ArcView GIS by ESRI. Additional data on land-use types, hydraulic soil properties and meteorological stations locations were included in the GIS data as well. Based on the mentioned data, internal landscape and soil hydraulic properties of the elementary basins were determined. A total of 1045 elementary basins were

5    outlined, with average area of 340 sq. km. The model forcing data were interpolated from 157 weather station data (see Fig. 1) to each elementary basin by the use of the inverse-distance method. Spatial distribution of soil, and land-use characteristics, as well as the calculated river network and elementary basins are shown in Fig 3.

In this study, the model was calibrated and validated against the Cheboksary reservoir daily water inflow observations beginning from January, 1, 1982 (the $1^{st}$ year after the reservoir filling to capacity) to December 31, 2016: calibration

10   covered the period of 2000-2010, the rest of the data were used for the model evaluation. The results of the model testing are presented in the next section.



a

b

c

d

**Figure 3 – Spatial data used in this study : soil type (a), land cover (b), stream network over topography (c) and sub-basin (d) maps.**



### 3.2  Multi-site weather generator (MSFR_WG)

The Multi-Site FRagment-based stochastic Weather Generator (MSFR_WG) is the stochastic model producing Monte-Carlo simulated time-series of daily weather variables (precipitation, air temperature and air humidity deficit), retaining statistical properties, both spatial and temporal, of the corresponding observed variables. The MSFR_WG's modeling procedure is based on so-called "spatial fragments' (SFR) resampling method" initially presented by Gelfan et al (2015). The SFR-method is a modification of the temporal fragments' (TFR) method proposed by Svanidze (1980) for stochastic simulation of highly autocorrelated time-series.

The SFR resampling method includes the following sequential steps:

1.  Computation of N normalized fields ("spatial fragments", SFR) of weather variables on the basis of available meteorological data. SFRs are computed for each of N years of observations by dividing each daily value of the specific variable on the corresponding spatially averaged annual value;

2.  Monte-Carlo simulation of synthetic time-series of $M$ spatially averaged annual weather variables reproducing temporal statistical features of the corresponding annual variables derived from observation data. Cross-correlation between annual values of the simulated weather variables is taken into account through the Cholesky's decomposition method.

3.  Calculation of synthetic daily fields of weather variables by multiplying the computed SFRs (see step 1) on the Monte-Carlo simulated spatially averaged annual value of the corresponding variables (see step 2). SRFs are randomly chosen from the available set by the Latin Hypercube method.

The advantage of MSFR_WG is that it contains small number of free parameters in comparison with widely-distributed spatial weather generators (see, for instance Khalili et al. (2011) and references therewith). The model does not require complex estimation procedures; this fact, hopefully, leads to increase of the model robustness. List of the MSFR_WG parameters as well as the results of the model testing are presented in the next section.

### 3.3 Forecasting procedure and verification metrics

The proposed forecasting procedure utilized in this study was verified by producing hindcasts of water inflow into the Cheboksary reservoir for 35 seasons in 1982 – 2016 from the 1$^{st}$ of April for 3 months ahead. For each i-th year of the verification period (i=1,2…,35), the procedure includes the following steps:

(1)  Spinup ECOMAG-based simulation ("warm start") using meteorological observation data several months prior to the forecast date in order to calculate the initial watershed hydrological state (soil, snow and channel water contents, groundwater level, soil freezing depth, etc.) that initializes the forecast;





(2) Selection of a weather scenario[1] from the ensemble $N_{EPS}$ of the observed weather or $N_{WG}$ of the generated weather for the forecast horizon;

(3) Calculation of the daily inflow hydrograph with the hydrological model driven by the assigned scenario for the forecast horizon;

(4) Repetitive selection of the next weather scenario (step 2) and calculation of the corresponding inflow hydrograph (step 3). Reselecting the scenarios ends with the last scenario in the ensemble and the corresponding ensemble of N (N =$N_{EPS}$ or N=$N_{WG}$) inflow hydrographs is formed;

(5) From each of the calculated time-series the following characteristics are derived: (1) inflow volume (hereafter referred to as *W*), (2) maximum inflow discharge (*Qmax*), (3) number of days with inflow discharge above mean observed discharge for the forecast horizon (*Nq*), (4) number of days with inflow discharge above mean maximum observed discharge for the forecast horizon (*NqMax*).

(6) Deterministic (ensemble mean) and probabilistic forecast are derived and verified for each of the inflow characteristics

The first five steps of the above procedure are illustrated by Fig. 4

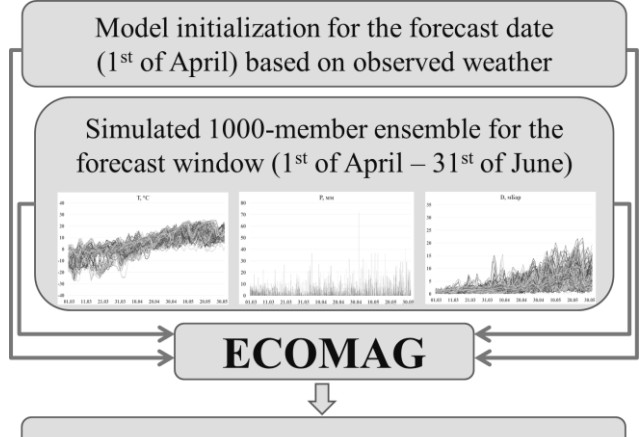

**Figure 4 – The proposed ensemble forecasting procedures**

The following forecast verification metrics were used to estimate the forecasts efficiency and skill. For the deterministic forecasts we used the estimates of mean, mean error (*ME*), *BIAS*, root-mean-squared error (*RMSE*), Pearson's correlation coefficient *R*. For categorical verification we used metrics that can be calculated with a contingency table, such as probability of detection (POD, shows the correct forecast fraction of the observed events), false alarm ratio (FAR, shows the

---

[1] Hereafter, by "weather scenario" we mean an array of weather time-series (daily precipitation amount, air temperature and humidity deficit) that are used to drive the hydrological model for the forecast horizon



fraction of forecasts that did not occur), frequency BIAS (shows correspondence of the observed and the forecasted events), Heidke skill score (HSS, shows the advantage of the forecast as compared to random forecast), Hanssen and Kuipers score (KSS, can detect if the forecast is hedging) and Symmetric Extremal Dependency Index (SEDI, evaluates the performance of the forecast of rare binary events). Detailed description of the used metrics has been presented, by Ferro and Stephenson, 2011, among others.

The probabilistic ensemble forecasts skill was assessed with several verification scores. For the binary probabilistic forecasts we used Brier score (BS) to assess the forecast error and Brier skill score (BSS) to compare it to the climatology (e.g. Murphy, 1973). The discrimination ability of the probabilistic forecast for events and non-events was demonstrated by the Relative Operating Characteristic curve and area under the curve (AOC, Mason and Graham, 1999). The Ranked Probability Score (RPS) and Ranked Probability Skill Score (RPSS) were used to evaluate the ability of the forecast to predict the category, in which the observed value fell into (see, for instance Franz and Hogue (2011) and references therewith).

## 4. Results and discussion

### 4.1 ECOMAG calibration and evaluation

The hydrological model was calibrated and validated against daily time-series of water inflow into the Cheboksary reservoir for the period of 1982 – 2016. The observed inflow data were derived from streamflow gauge measurements on major Cheboksary reservoir tributaries and does not account for inflow from the upstream Nizhegorodskoe reservoir.

The model performance was estimated by Nash-Sutcliffe efficiency (NSE) criterion for daily streamflow discharge and by Pearson's correlation coefficient between observed and simulated runoff volume for 3 months (April to June), as well as between observed and simulated maximum streamflow discharge during the same period. Fig. 5 shows the compared hydrographs of observed and simulated daily inflow discharges. Table 1 summarizes the obtained results. One can see from these illustrations that the modeling results have shown good agreement with the observations.

**Table 1. ECOMAG calibration and evaluation results**

|  | Calibration (2000 – 2010) | Evaluation (1982 – 1999; 2011 – 2016) | Whole period (1982 – 2016) |
|---|---|---|---|
| $NSE$ | 0.83 | 0.79 | 0.80 |
| $R_{vol}$ | 0.93 | 0.81 | 0.84 |
| $R_{max}$ | 0.94 | 0.75 | 0.77 |

NSE – Nash-Sutcliffe criterion, $R_{vol}$, $R_{max}$ – correlation coefficients for inflow volume in April – June and maximum inflow discharge values in the same period

Figure 6 shows scatterplots of the observed and simulated characteristics of the inflow into the Cheboksary reservoir in April – June. In general, the inflow volume is well simulated, yet slightly underestimated for the high flows (above 50 km³, see Fig. 6, upper left). Maximum inflow discharge is a highly uncertain characteristic, but is still well simulated by the model




(see Fig. 6, lower left). The number of days above a certain inflow discharge threshold is a highly important characteristic for various users, e.g. waterways navigation and water supply. For the number of days above long-term (1982-2016) mean inflow discharge during the period between April and June, the model shows less days than the observed ones – 31 against 36 (see Fig. 6, upper right). For the number of days above long-term mean maximum inflow discharge the model also shows

5    less days – 13 modelled against 17 observed (see Fig. 6, lower right).



**Figure 5 – Time-series of observed and simulated hydrographs of inflow into the Cheboksary reservoir**

Bias of the inflow volume in April – June for the whole period 1982 – 2016 is -0.84 km$^3$; RMSE of the inflow volume (6.15 km$^3$) appeared one third less than the observed data standard deviation (9.41 km$^3$). Bias of the maximum inflow

10    discharge is -0.95 m$^3$/s; RMSE – 2321 m$^3$/s, which is also lower than the observed standard deviation of the maximum inflow discharge (3385 m$^3$/s).




Magnitude of the used metrics and error estimation has led to an assumption that the model is suitable to act as a core component of the long-term hydrological forecasting system.

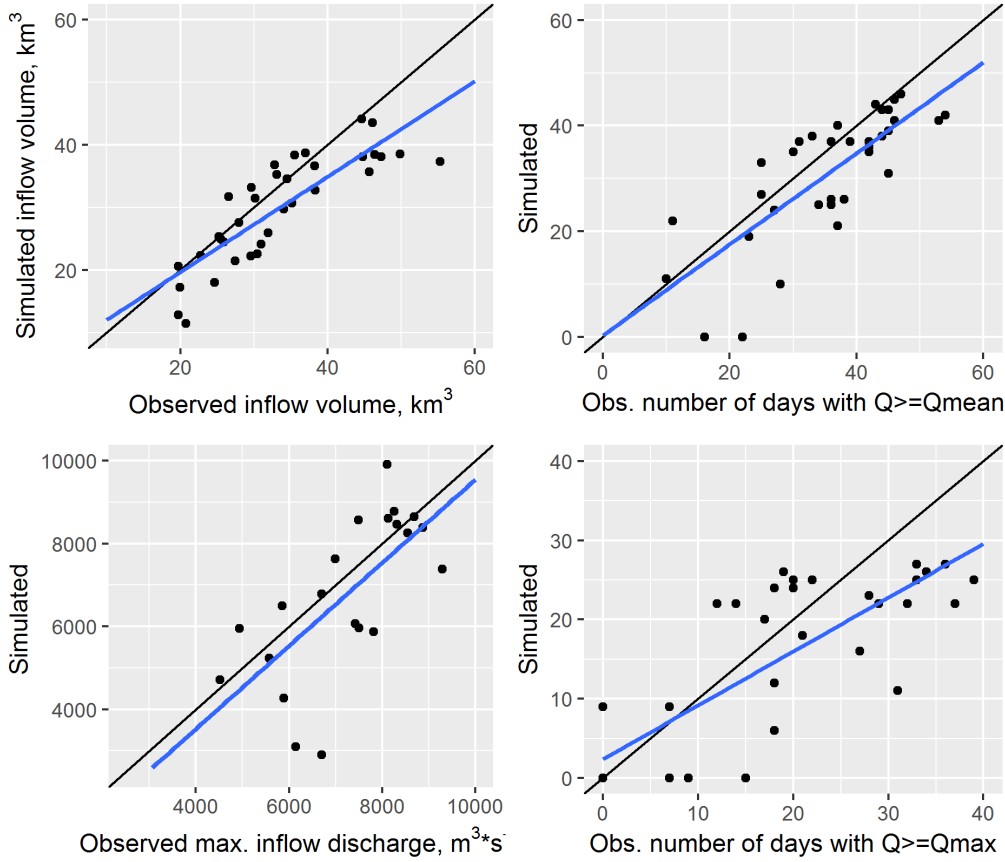

**Figure 6 – Model performance scatterplots for the observed and simulated inflow characteristics during April – June: total inflow volume (upper left), maximum discharge (lower left), number of days above mean inflow discharge (upper right), number of days above mean maximum inflow discharge (lower right)**

**4.2 MSFR_WG: parameter estimation and model testing**

Time series of daily precipitation, air temperature and humidity deficit observed in the meteorological stations located at the Cheboksary reservoir basin for 51 years (1966-2016) are used to estimate nine parameters of the developed stochastic model. The parameters (see Table 2) have been estimated by the method of moments. The stochastic models were comprehensively tested through its ability to reproduce the main statistical characteristics of meteorological processes at the Cheboksary reservoir basin. For testing, we compared only those characteristics of the observed and simulated time-series which are neither the parameters of the model nor a single-valued function of the parameters as suggested in (Gelfan, 2010). Statistics of the 1000-year Monte-Carlo generated time-series of daily meteorological variables were compared with the following corresponding statistics derived from observations: mean and variation of annual and monthly values,



autocorrelation functions of daily and monthly values of the specific variables. Results demonstrating comparison between statistical properties of the observed and simulated series are shown in Supplementary Materials for spring months and for 15 selected stations (Figs. 1-5).

**Table 2 – List of the MSFR_WG parameters**

| | Parameter | Estimate |
|---|---|---|
| 1 | Spatially averaged annual mean of air temperature, $^{O}$C | 4.58 |
| 2 | Spatially averaged annual mean of precipitation intensity, mm/day | 1.52 |
| 3 | Spatially averaged annual mean of air humidity deficit, mb | 3.15 |
| 4 | Standard deviation of spatially averaged annual air temperature, $^{O}$C | 0.99 |
| 5 | Standard deviation of spatially averaged annual precipitation, mm/day | 0.18 |
| 6 | Standard deviation of spatially averaged annual air humidity deficit, mb | 0.69 |
| 7 | Pearson coefficient of correlation between spatially averaged annual temperature and precipitation | 0.02 |
| 8 | Pearson coefficient of correlation between spatially averaged annual air humidity deficit and precipitation | -0.51 |
| 9 | Pearson coefficient of correlation between spatially averaged annual air humidity deficit and temperature | 0.29 |

### 4.3 Ensemble forecast verification

#### 4.3.1 Deterministic forecast

For the verification purposes, ensembles of the forecasted inflow characteristics were averaged to produce a single-value hindcast of the desired characteristic – inflow volume ($W$), maximum inflow discharge ($Qmax$), number of days with inflow

10    discharge above mean observed discharge ($Nq$), number of days with inflow discharge above mean maximum observed discharge ($NqMax$). Values of all hindcasts were treated as continuous variables to assess the properties of the forecast: mean, standard deviation, mean error, bias, RMSE and Pearson's correlation coefficient.

**Table 3 – Deterministic hindcasts assessment values (the designations used are described in the text)**

| Inflow characteristics | Observed | Mean | | Mean error | | BIAS | | RMSE | |
|---|---|---|---|---|---|---|---|---|---|
| | | ESP | WG | ESP | WG | ESP | WG | ESP | WG |
| **W** | 33.38 | 32.34 | 33.54 | 1.04 | -0.17 | -3% | 1% | 5.33 | 5.69 |
| **Qmax** | 9355 | 9463 | 9958 | -108 | -603 | 1% | 6% | 1970 | 2244 |
| **Nq** | 35.9 | 35.92 | 36.10 | 0.02 | -0.15 | 0% | 0% | 7.97 | 8.81 |
| **Nmax** | 17.0 | 16.17 | 17.09 | 0.86 | -0.06 | -5% | 0% | 7.38 | 8.19 |

15    Fig. 7 displays the Taylor diagram (Taylor, 2001) of the hindcast assessment. The values of all characteristics are normalized by dividing the RMS and the standard deviations of the forecasts by the standard deviation of the observations. It can be seen that the ESP-based hindcasts of $W$, $Qmax$ and $NqMax$ are slightly better correlated with the observations than the WG-based





hindcasts. Pearson's $R$ values of the ESP-based forecasts are over 0.8 for all characteristics, except for $Nq$. Hindcasts of $Nq$ are less correlated with the observations, with value of $R$ for ESP-based of 0.73 and WG-based – as low as 0.63. Hindcasts of $Qmax$ and $NqMax$ show RMSE values around 0.55 – 0.65 fraction of the standard deviation of the corresponding observed characteristics.

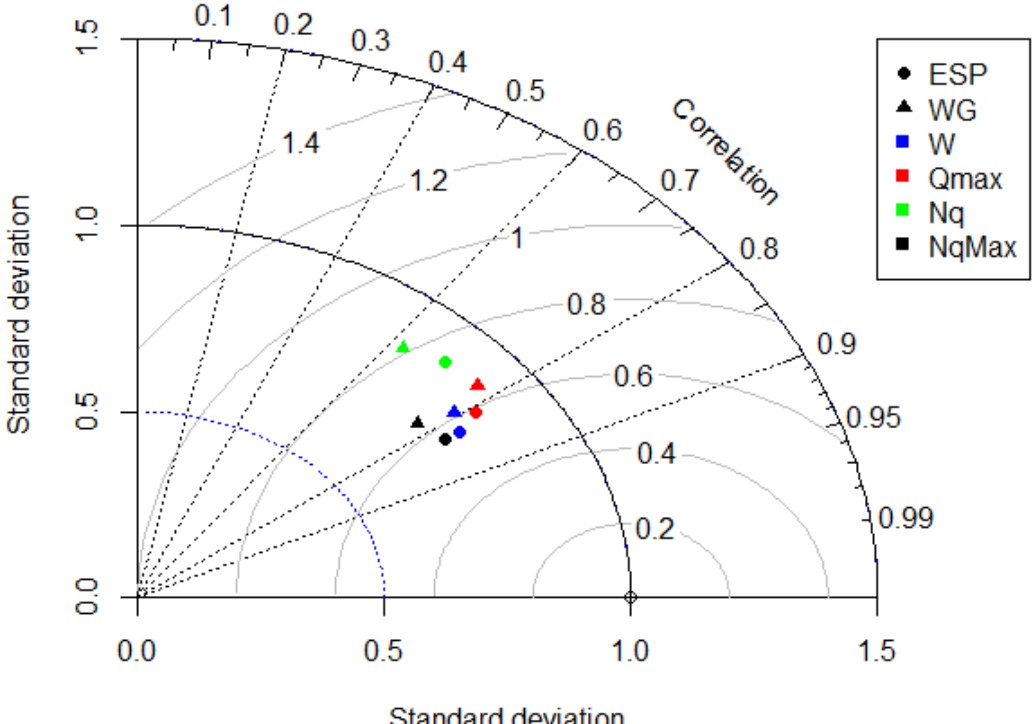

**Figure 7 – Taylor diagram of ESP (circle) and WG-based (triangle) forecasts of inflow volume (in blue), maximum inflow discharge (red), number of days with inflow discharge above mean (green), number of days with inflow discharge above maximum (black).**

For the purposes of reservoir management, it is often crucial to determine whether the expected inflow will exceed the annual mean value. To verify the employed methodology for capabilities of predicting this exceedance, observations and hindcasts were turned into binary vectors with "zero" value representing the event of non-exceedance of the mean annual value and "one" for the event occurrence. For $W$ the event occurs with the exceedance of mean inflow volume during April – June, for $Qmax$ – with the exceedance of maximum inflow discharge, for $Nq$ – the mean number of days with inflow discharge above mean value, for $NqMax$ – the number of days with inflow discharge exceeding the mean maximum value. Binary metrics of the hindcasts were then assessed with the use of contingency tables. Assessed binary verification metrics are shown in Table 3.

The hindcasts of $Qmax$ show perfect detection estimates, but this seems to be an outcome of hedging, as the frequency Bias is very high, which means that both forecast methodologies tend to overpredict $Qmax$, so with the high values of False



Alarm Ratio and Hansen-Kuipers score. However, the forecasts still are significantly better than the forecast by chance with Heidke Skill Score of more than 60%, and are capable of detecting the occurrence of rare extreme events, which is shown by high values of Symmetric Extremal Dependency Index.

**Table 3 – Verifications metrics for the binary forecasts**

| | POD | | FAR | | Freq. BIAS | | HSS | | KSS | | SEDI | |
|---|---|---|---|---|---|---|---|---|---|---|---|---|
| | $R^* \in [0;1]$; | | $R \in [0;1]$; | | $R \in (-\infty;+\infty)$ | | $R \in (-\infty;1]$; | | $R \in (-1;1)$ | | $R \in (-1;1)$ | |
| Inflow | PFM$^{**}$=1 | | PFM=0 | | PFM=0 | | PFM=1 | | PFM=1 | | PFM=1 | |
| characteristics | ESP | WG | ESP | WG | ESP | WG | ESP | WG | ESP | WG | ESP | WG |
| **W** | 0.87 | 0.87 | 0.24 | 0.32 | 1.13 | 1.27 | 0.66 | 0.55 | 0.67 | 0.57 | 0.82 | 0.73 |
| **Qmax** | 1.00 | 1.00 | 0.33 | 0.37 | 1.50 | 1.58 | 0.66 | 0.61 | 0.74 | 0.70 | 0.93 | 0.92 |
| **Nq** | 0.77 | 0.73 | 0.23 | 0.20 | 1.00 | 0.91 | 0.39 | 0.41 | 0.39 | 0.42 | 0.53 | 0.57 |
| **Nmax** | 0.79 | 0.73 | 0.17 | 0.20 | 0.95 | 0.91 | 0.60 | 0.41 | 0.60 | 0.42 | 0.76 | 0.57 |

$^*$R is the Range of the metric value

$^{**}$ PFM is the Perfect Forecast Metric value

### 4.3.2 Probabilistic forecast

The ensemble forecast can be used to assess the probability of an event occurrence, similar to as described in previous section. For this purpose the observations are left in binary form, and the forecast represents event probability in [0, 1]. We

used Brier score to assess the forecast performance and Brier skill score to compare the forecast to climatology. The results are shown in Table 4. WG-based forecast of both *W* and *Qmax* have less errors and show more skill compared to sample climatology.

**Table 4 – Brier scores (BS) and skill scores (BSS) for the forecast of *W* and *Qmax***

| | BS | | BSS | |
|---|---|---|---|---|
| **Forecast** | $R^* \in [0;1]$; PFM$^{**}$=0 | | $R \in [0;1]$; PFM=1 | |
| | **W** | **Qmax** | **W** | **Qmax** |
| **ESP** | 0.13 | 0.16 | 0.28 | 0.48 |
| **WG** | 0.10 | 0.12 | 0.45 | 0.59 |

$^*$R is the Range of the metric value

$^{**}$ PFM is the Perfect Forecast Metric value

The ability of probabilistic forecast to distinguish between an event and non-event of exceedance of the mean inflow volume in April – June was further tested with plotting a relative operating characteristic curve that shows the forecast's false alarm rate against probability of detection over multiple probability thresholds. Area under curve (AOC) for the forecast of *W* for




both WG-based and ESP-based forecasts show good result at around 0.92 – 0.94, for the forecast of *Qmax* both approaches show AOC of 0.93.

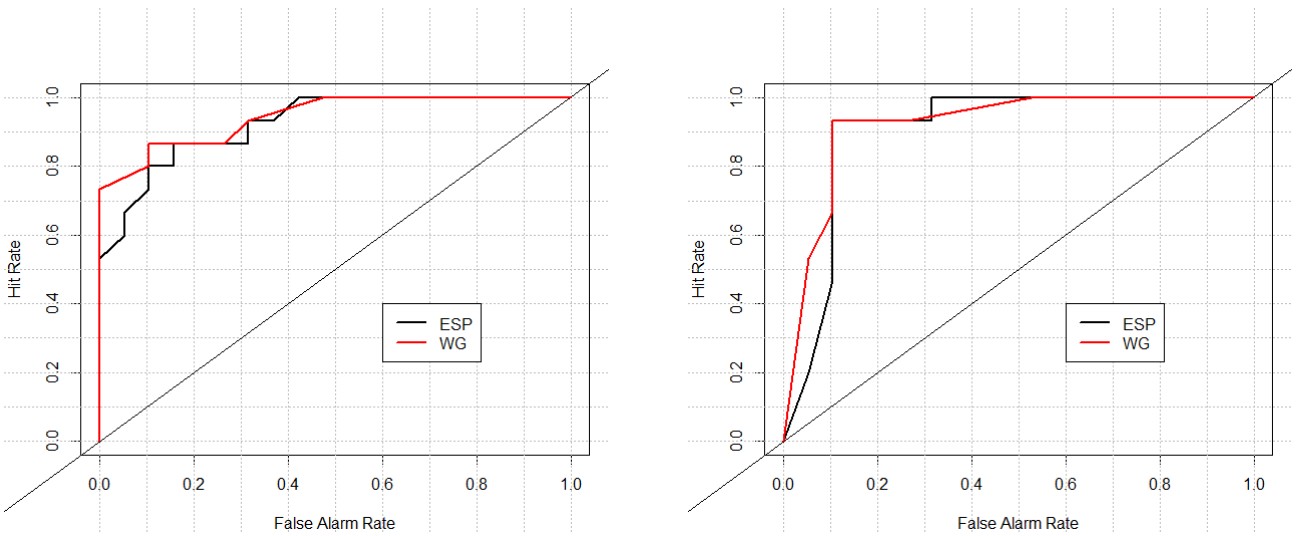

**Figure 8 – Relative operating characteristic curves for W (left) and Qmax (right) forecasts**

One of the main advantages of the ensemble forecasting is the ability to assess the uncertainty that is nested in the future
possible behavior of the hydrological system. The resulting ensemble is used to create cumulative distribution function (CDFs) of the desired characteristic in $j^{th}$ hindcast as

$$F_m(j) = \sum_{i=1}^{m} f_i(j), \; m=1,\dots,M; j=1,\dots,N, \qquad (1)$$

where $M$ are the forecast probability bins on the interval [0;1]; $N$ – total number of hindcasts; $f_i$ – probability of forecast in $m^{th}$ bin. Observations for this purpose are treated as Heaviside step function.

CDFs of the forecasted inflow volume W for the period from April, 1 to June, 30 of 35 years (1982-2016) are shown in Fig. 9. Three CDFs are combined in each plot: two CDFs of forecasts calculated under ESP-based and WG-based weather scenarios, and CDF of the observed inflow volume in the specific year. One can see from Fig. 9 that for the most of the years the inflow is not far from the most probable one, in other words, CDF of the forecasts crosses CDF of observations around 50% probability. For almost all years observed inflow lies within the range of the ensemble. Exceptions are 1994, 2002,

2005, 2011, i.e. once per 8-9 years, on the average, ensemble forecast range does not cover observed inflow because of large forecast errors.



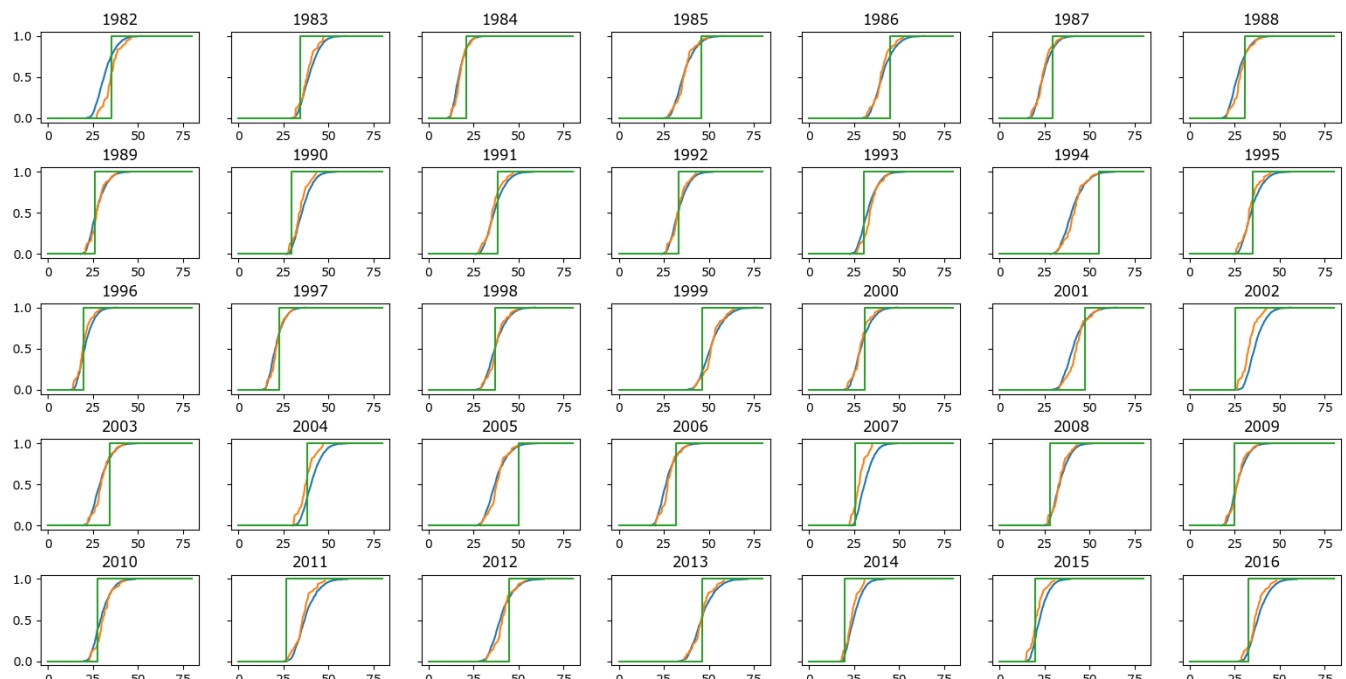

**Figure 9 – Cumulative probability distribution functions for *W* in April – June for all years between 1982 and 2016. Green line – observed inflow, orange – ESP-based forecast, blue – WG-based forecast**

The ability of forecast to correctly predict the category of the event occurred within *M* categories was measured by Ranked Probability Score (RPS). It can also be treated as the mean squared error of the probabilistic forecast. Given M event categories the RPS is computed for each hindcast as

$$\overline{RPS} = \sum_{m=1}^{M} MSE_m = \frac{1}{N} \sum_{j=1}^{N} \sum_{m=1}^{M} \left( F_m(j) - O_m(j) \right)^2 \quad , \tag{2}$$

where Fm(j) is the predicted probability of the event to occur in the category m, Om(j) is the observed event probability value. Om(j) is presented by Heaviside step function, equal to 0 in the non-event category and turning to 1 if the event value is exceeded. The forecast efficiency can be measured by Ranked Probability Skill Score, where the verified forecast is related to climatology:

$$RPSS = 1 - \frac{\overline{RPS}}{\overline{RPS_{cl}}} \quad , \tag{3}$$

where $\overline{RPS}$ - is the metric value over multiple forecasts, $\overline{RPS_{cl}}$ - ranked probability score of the climatology. RPS can range from 0 to 1 with the perfect forecast score of 0. RPSS is defined in the interval (-∞; 1], 0 indicates that the forecast has no skill against the climatology, perfect forecast is 1.

9999




The ESP-based and the WG-based ensembles show fairly good results in predicting the category of the observed value (see Table 5). For ESP forecast the value of RPS for the *W* forecasts are slightly better than of the WG forecasts. Both forecasts demonstrate efficiency against the climatology, according to the RPSS value, around 30% on average both for *W* and for

*Qmax*.

**Table 5. Ranked Probability Score and Skill score for the forecasts**

| Inflow characteristics | RPS ESP | RPSS ESP | RPS WG | RPSS WG | RPS clim |
|---|---|---|---|---|---|
| *W* | 0.054 | 0.375 | 0.063 | 0.278 | 0.087 |
| *Q max* | 0.072 | 0.284 | 0.081 | 0.195 | 0.101 |

It can be seen from Fig. 9 that the CDFs have appeared to be close to each other for both types of weather scenarios for almost all hindcasts. However, as mentioned above, the sample variance of the CDF may be significantly different due to

different amount of scenarios in the ensembles: 51 in the ESP-based ensemble and 1000 in the WG-based ensemble. To illustrate such difference in Fig. 10-11 we show confidence bands for CDFs derived from both forecasting approaches. A two sided confidence band can be expressed from Dvoretzky-Kiefer-Wolfowitz inequality (e.g. Massart, 1990):

$$P\left(\sup_{x \in R}\left|\hat{F}_n(x) - F(x)\right| \geq \varepsilon\right) \leq 2\exp\left(-2n\varepsilon^2\right) \tag{4}$$

where $F(x)$ and $\hat{F}_n(x)$ are the CDF's ordinate and its empirical estimation from a sample of size *n*, respectively; $\varepsilon$ is the

constant depending on the confidence probability α as

$$\varepsilon = \sqrt{\frac{1}{2n}\ln\left(\frac{2}{1-\alpha}\right)} \tag{5}$$

For the pre-assigned confidence probability α, the upper ($U(x)$) and the lower ($L(x)$) confidence bands of the empirical CDF $\hat{F}_n(x)$ are defined from (5) and (6) as:

$$U(x) = \min\left[\hat{F}(x) + \varepsilon, 1\right]$$

$$L(x) = \max\left[\hat{F}(x) - \varepsilon, 0\right] \tag{6}$$

Figs. 10-11 clearly demonstrate the magnitude of sample variance of the ESP-based forecast as compared to the WG-based forecast. We assume that this difference should be taken into account by the ensemble forecast developers when they use statistical verification metrics for assessment of forecast performance, as well as by the users when they interpret the forecast.



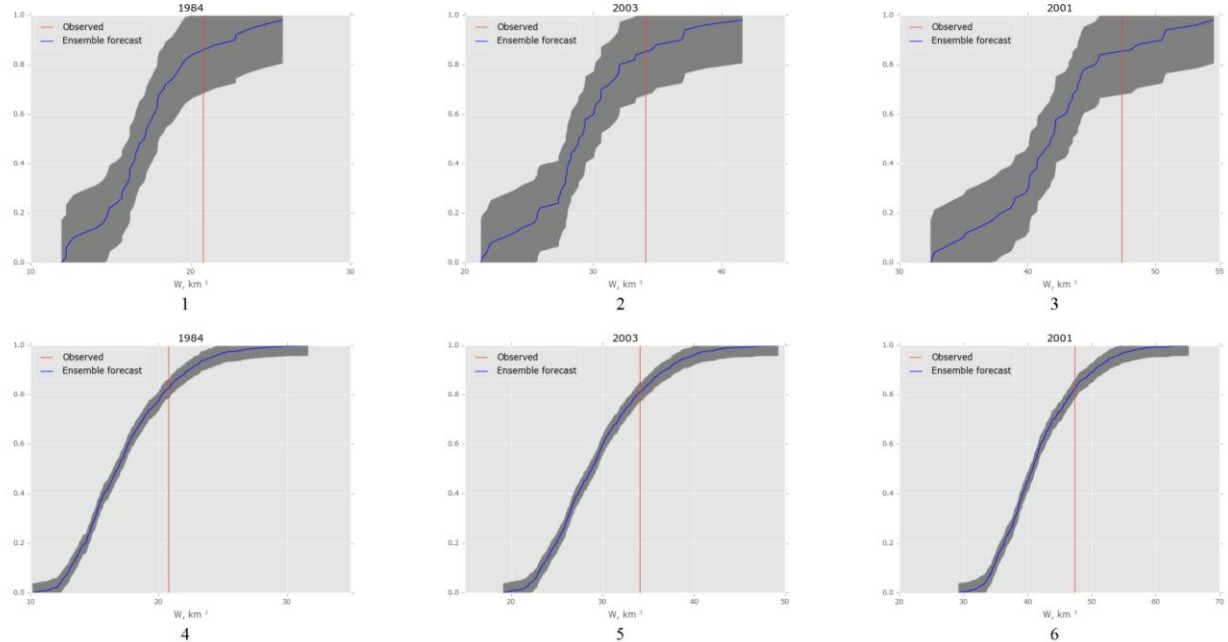

**Figure 10 – Cumulative probability distribution functions for *W* in April – June for selected years between 1982 and 2016 for ESP-based forecast (1 – 3) and WG-based forecast (4 – 6)**

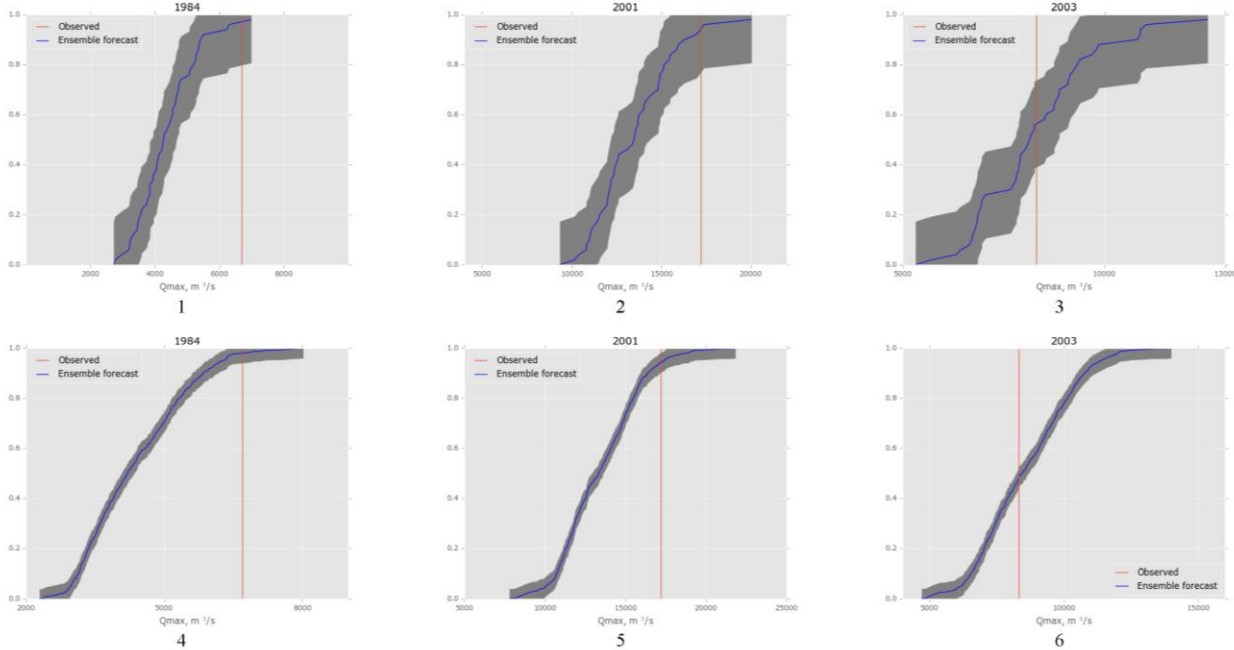

5    **Figure 11 – Cumulative probability distribution functions for *Qmax* in April – June for selected years between 1982 and 2016 for ESP-based forecast (1 – 3) and WG-based forecast (4 – 6)**



## 4.4 Influence of the forecast issue date on the forecast performance: "forecastability maps"

Numerical experiments were carried out to estimate sensitivity of the forecast performance (in terms of NSE-criterion) to changes in the forecast horizon and issue date. The experiments were designed as follows.

On the first step, the procedure shown in Fig. 4 was applied on the forecast issue date (ID) of March 1st (ID$^{(1)}$=01/03). for

the lead-time (LT) of 1 month (LT$^{(1)}$=1), i.e. an ensemble of inflow hydrographs was calculated staring from March 1st to March 31st for each year of the hindcasts (35 years in 1982 – 2016). From the calculated ensemble, four inflow characteristics (W, Qmax, Nq, NqMax) were estimated, compared to the corresponding observed characteristics and NSE criterions was calculated for each characteristic (NSE$_W$, NSE$_{Qmax}$, NSE$_{Nq}$, NSE$_{Nmax}$, respectively). As a result, after the 1$^{st}$ step of the experiment, ordinates of two-dimension functions NSE$_W$(ID,LT), NSE$_{Qmax}$(ID,LT), NSE$_{Nq}$(ID,LT), and

NSE$_{Nmax}$(ID,LT) were estimated at the point (ID$^{(1)}$,LT$^{(1)}$) of the space (ID,LT).

On the second step of the experiment the forecast issue date was shifted 10 days forward (ID$^{(2)}$=10/03), and the procedure was repeated for the same lead-time (LT$^{(1)}$=1), i.e. the forecast was issued for the period of March 10th – April 9th. As a result, ordinates of the above functions were estimated on the point (ID$^{(2)}$,LT$^{(1)}$).

Then, the procedure was repeated 7 times for the issue dates ID$^{(3)}$=20/03, ID$^{(4)}$=01/04… ID$^{(9)}$=20/05 and periods from March

20th to April 19th, April 1st to April 30th, …, May 20th to June 19th, respectively.

On the next step, the issue date was shifted back to March 1st (ID$^{(1)}$=01/03), but the lead-time was set to 2 months (LT$^{(2)}$=2), i.e.. for April 1st to April 30th. The experiment finished with all 27 combinations of ID$^{(i)}$ (i=1,2,…9) and LT$^{(j)}$ (j=1,2,3). As a result, four matrixes (9x3) were built: NSE$_W$(ID$^{(i)}$,LT$^{(j)}$), NSE$_{Qmax}$(ID$^{(i)}$,LT$^{(j)}$), NSE$_{Nq}$(ID$^{(i)}$,LT$^{(j)}$), NSE$_{Nmax}$(ID$^{(i)}$,LT$^{(j)}$)

Using the obtained data, 2D surfaces illustrating the forecast efficiency dependence on ID and LT for each of the desired

characteristics were developed. We propose to name these surfaces as "forecastability maps". The term "forecastability " is similar to meteorological term "effective predictability" (Douville, 2010), which depends, among other, on subjective factors, including the type of forecasting methodology, experience of the forecaster, observational network features, initial data content and quality, etc. We use this term to emphasize the difference of "forecastability" from "predictability", the latter being an objective physical feature of the hydrological system and does not depend on the forecasting methodology

(see e.g. Report…, 2002; Shukla et al., 2013).

Forecastability maps of four inflow characteristics are shown in Fig. 12. (The maps are shown only for the ESP-based ensembles). Blank space in the upper right corner of forecastability maps for *Nq* and *NqMax* (Fig. 12c, d) is due to the zero values of variance of these characteristics at the end of April and May, so NSE function does not exist in this area. Analysis of the forecastability maps has shown that maximum forecast efficiency is achieved for forecasts issued in the beginning of

30 April with lead-time up to 2 months. Unsurprisingly, the broadest area of efficient forecasts was obtained for the forecast of W while the narrowest area was obtained for *Nq* and NqMax. We denote that all forecasts retain efficiency (*NSE* > 0) for all issue dates and lead-times used in the experiment (except the mentioned above spaces for *Nq* and *NqMax*). However, the forecasts in the beginning of spring appear to be more efficient than in May.



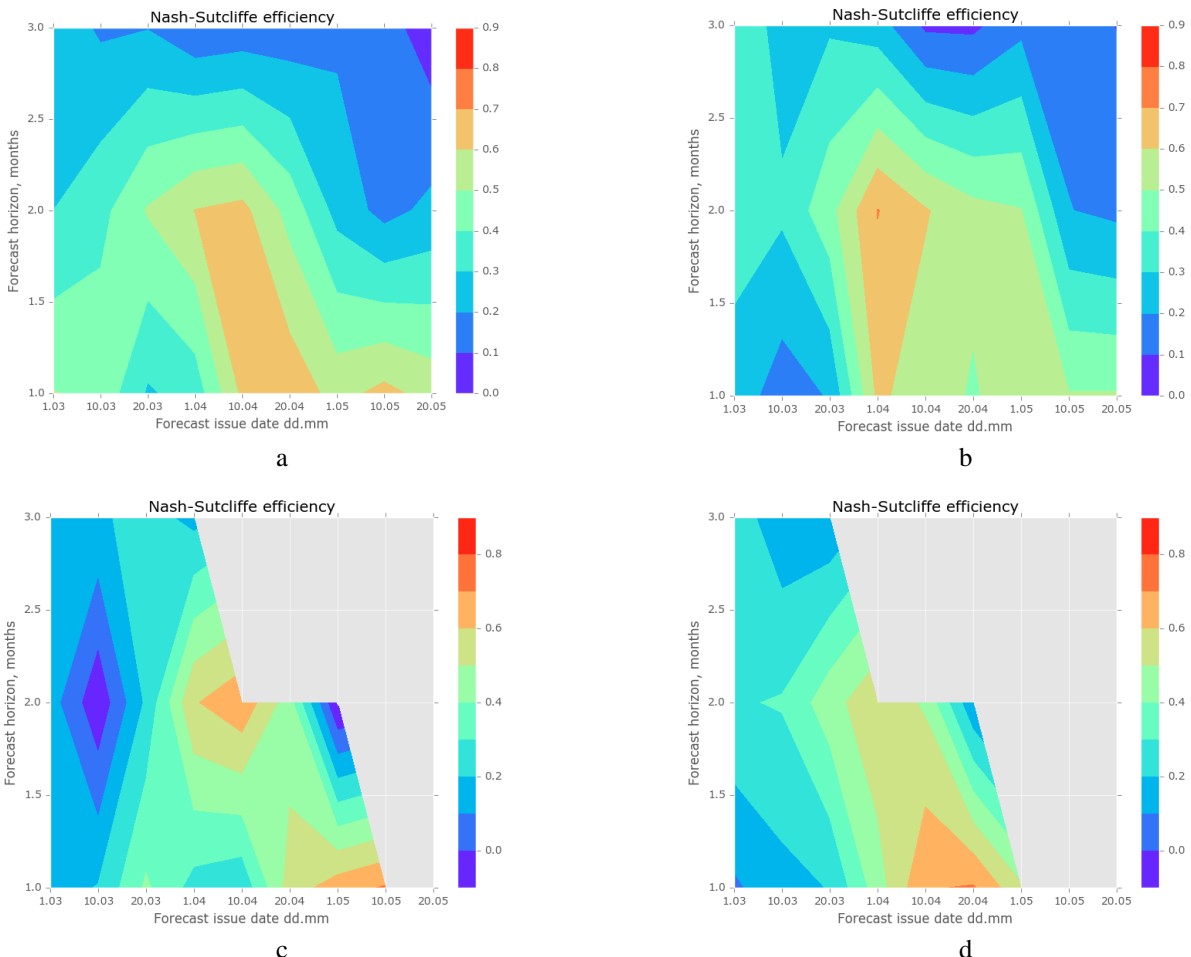

**Figure 12. Forecastability maps of *W* (a), *Qmax* (b), *Nq* (c) and *NqMax* (d)**

## 4.5 Use of the developed methodology for operational forecast: freshet of 2017

Weather conditions during the spring freshet formation in 2017 in the Cheboksary river basin were significantly different
from the long-term average conditions during the snowmelt period. The initial conditions of the basin before the snowmelt
onset were close to the average: snow water storage was 10 to 15% above the average, soil water storage and freeze depth –
close to the average values. However, anomalous warm and sunny weather that settled over the basin in the first half of
March has led to commencement of snowmelt and river stage ascent at least half of the month earlier than the usual long-
term average dates. The last decade of March was, on the contrary, cold and damp and the precipitation amount has exceeded
two-thirds of the annual average (20.7 mm over 30.6 mm on the average). As a result, by the end of March the inflow
volume into the reservoir has exceeded the annual average (5.13 over 3.89 km$^3$ on the average). Periods of intense snowmelt,
interchanged with cold spells and large amount of precipitation, including snowfall, have been observed during April and





May in 2017 (a number of stations have registered snowfall even in June). Such diversity in weather conditions of runoff formation during the snowmelt period and their difference from the average conditions has influenced on the regime of inflow into the Cheboksary reservoir. Fig. 13 shows the difference of daily inflow during March – June 2017 from the climatic mean. Given these conditions, the forecast errors may have significantly increased.

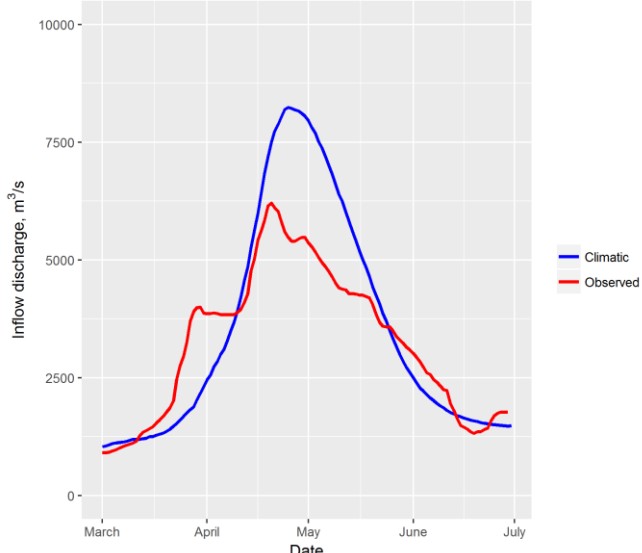

**Figure 13. Daily inflow into the Cheboksary reservoir during March – June 2017 compared to the climatic mean inflow**

The developed forecast approach was tested in operational mode during the freshet period of 2017. The forecasts were issued on March $1^{st}$, $15^{th}$ and $27^{th}$ for the period between April and June. Fig. 14 shows daily forecast ensembles for April $1^{st}$ – June $30^{th}$ compared to the observed inflow data (the ESP-based weather scenarios have been used to calculate the represented

10   hydrographs).

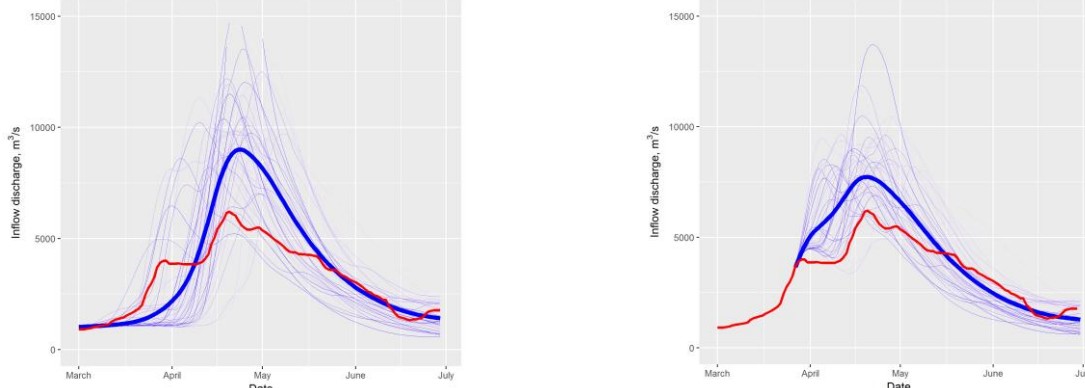

**Figure 14. Daily forecast ensembles (in blue) for April – June issued on March $1^{st}$ (left) and March $27^{th}$ (right). Bold blue line – ensemble average, bold red line – observed inflow discharge.**


Box plots of the forecasts are given in Fig 15. All forecasts of *W* have shown low errors (Fig. 15a), unlike the *Qmax* forecast (Fig. 15b). However, the forecast distribution grasps the observed maximum inflow discharge volume. The forecast of *N* has shown low error (Fig. 15c). However, the *Qmax* forecast has overestimated the amount of days with extreme discharge, which was not observed in 2017 (Fig. 15d).

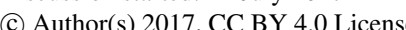

**Figure 15. Box plots of ensemble forecasts of W (a), Qmax (b), N (c) and Nmax (d) for April – June 2017**



## 5 Conclusions

Accumulation and retention of water in the surface and subsurface storages of the river basin during several months of negative air temperature allows for long-term (sub-seasonal to seasonal) forecasting of the spring freshet runoff in cold regions. In the last decades there has been a significant evolution in the methodology of long-term hydrological forecasting, from simple regression deterministic approaches to ensemble forecasting, based on distributed hydrological models, describing physical mechanisms of runoff generation and capable to assimilate new observations as well as weather forecasts both in deterministic and probabilistic format. Scientific and technical prospects have led to improvements in spatial and temporal resolution of the freshet runoff volume forecast, its accuracy and lead-time. Additionally, on the basis of process-oriented hydrological models, an opportunuty appears for long-term forecasting of the internal hydrological variables other than runoff (snow, soil moisture, etc.) (see comprehensive review presented by Koster et al., 2017). At the same time, despite significant achievements in long-term forecasting methodology, the forecasts are still far from perfection and implementation of the state-of-the-art methods into practice should be carried out with extensive testing.

The paper describes the development and preliminary assessment results of the long-term forecasting methodology for the water inflow into the Cheboksary reservoir – one of the eleven major river reservoirs of the Volga-Kama Reservoir Cascade. The methodology is based on a combination of semi-distributed hydrological model ECOMAG that allows for ensemble calculation of inflow hydrographs with two different sets of weather ensembles for the lead-time period: (1) observed weather data constructed on the basis of the ESP methodology and (2) synthetic weather data simulated by a weather generator. Compared to deterministic forecasts of inflow into reservoirs, that are used in common practice in Russia, the proposed methodology allows for:

1. Calculation of initial (up to forecast issue date) state of the river basin (snow water storage, river network storage, soil moisture content and freeze depth) based on common weather data. This appears to be a significant advantage of the system, given the snow and soil water storage observational network shrinkage and absence of soil moisture observations during the wintertime.

2. Forecasting extended list of river flow characteristics (maximum inflow discharge, duration of periods with flow above threshold etc.), which enhances the information content of a forecast.

3. Forecasting is available from any given date, e.g. different lead-times are available within a single forecast methodology. The existing long-term forecasting algorithms differ in dependence on the forecast horizon, demand for initial data, parameter set etc.

The main findings of our research are the following:

1. Two different forecast ensembles (ESP-based and WG-based) were developed for the lead-time period and have shown a significant similarity with each other. Quality of the inflow forecasts has appeared to be slightly better with the ESP-based ensemble, but the detected differences are not significant. Moreover, the WG-based probabilistic forecast shows a significantly lower sample variance than the ESP-based forecast. As a result, confidence intervals





of CDF of the ESP-based forecasts are much wider than the corresponding interval of the WG-based forecast CDF. This problem may generate large sample variance of the ESP-based forecast verification metrics, and consequently, difficulties with the forecast interpretation by a decision maker.

2.  A methodology was proposed for forecast efficiency assessment by the "forecastability maps" analysis – a 2-dimensional image of forecast quality depending on its issue date and lead-time. The "forecastabiliy map" analysis showed that the maximum efficiency of the forecast of all characteristics of water inflow into the Cheboksary reservoir can be achieved by issuing the forecast with the presented methodology in the beginning of April with 2 months lead-time. The broadest area of efficient forecasts, as expected, was for the forecast of inflow volume, the narrowest – for duration of high-flow spells. The forecasts retain efficiency (NSE>0) for all dates and all lead-times within the selected period, however, forecasts issued in the early spring are more efficient than in May.

3.  The developed methodology was tested by 35 years of hindcasts and demonstrates satisfactory performance. Additionally, the methodology was succesfully applied for operational forecast of the just ended freshet of 2017.whose conditions were significantly different from typical conditions

## 6 Acknowledgements

The research related to developing methods of ensemble forecast and forecast verification technique was financially supported by the Russian Foundation for Basic Researches (grants No. 16-05-00679 and 16-05-00599). Other researches, including that related to assessing sensitivity of the forecast results to the forecast issue date, were financially supported by the Russian Science Foundation (grant No. 14-17-00700).

The present work was carried out within the framework of the Panta Rhei Research Initiative of the International Association of Hydrological Sciences (IAHS).

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
