# Peer review of "Long-term ensemble forecast of snowmelt inflow into the Cheboksarv reservoir under the two different weather scenarios"

_Hydrology and Earth System Sciences, 2017_

## Referee Comment (RC1) · Anonymous Referee #1 · 14 Aug 2017

The research presented in this manuscript compares two methods for producing longterm hydrological forecasts: Extendred Streamflow Prediction and forecasts in which the metheorological scenarios are based on a weather generator. In both cases, the hydrological model is ECOMAG, a semi-distributed process-based model. The methods are applied on an interesting case study, which is a reservoir that is fed by a very large snow dominated catchment. Unfortunately, in my opinion the manuscript would require many improvements regarding both content and presentation before being published in HESS. I would personally advise that it be rejected in its present form, even if I think that the general objective of implementing a long-term hydrological forecasting system for the Cheboksary reservoir is relevant and interesting.

In the following, I will explain my opinion in greater detail. I really appreciate the effort of the authors and in no way I wish to discourage them. On the contrary, I sincerely hope that this discussion can be useful for them.

**MAJOR COMMENTS**

1. The original contribution is not clear/significant enough

Extended streamflow prediction (ESP) was proposed by Day in 1985 and is very wellknown. In consequence, this is not an original element of the research presented herein. The specific weather generator used in the research was previously developed in another study, so that also does not represent a strong original content. The same consideration applies to the hydrological model ECOMAG. While the authors clearly state the problematic in their introduction (finding long-term predictability for snow-dominated catchments, in particular large multipurpose reservoirs), their contribution is not highlighted.

The three types of long-term forecasting systems currently operational on the VKRC system, described on page 2 lines 8-17, are never compared to the ESP and WG-based forecasts proposed by the authors. In my opinion, this is very unfortunate. As mentioned above, ESP is not a new technique. However, it would be interesting to compare it to a variety of other techniques, more or less sophisticated. I also think it would help the reader to appreciate your contribution. In fact, many of the results presented in the manuscript do not appear to me as a clear improvement over climatology. For instance, in Table 5, the skill scores obtained by the WG-based forecasts are all below 0.5. While I do agree that this represents an improvement over climatology, it is not a large one. I don't have any problem with this (a slight improvement over climatology), but it would probably be much more convincing to see (also) the improvement relative to at least one of the current operational methods mentioned on page 3. Figures 13 and 14 also support my comment: the forecasts presented on Figure 14 appear only slightly different from the climatology (please also see comment 2 about climatology)
presented on Figure 13. This is especially true for the forecasts issued on March 1st (Figure 14 left) compared to Figure 13.

To sum up my first major comment: I think the authors should clarify and emphasize their original contribution, which is not clear for me at the moment. One way to do so, besides phrasing it more clearly, is to include a comparison with at least one of the 3 operational forecasting systems described on page 2.

2. Some methodological/conceptual elements need clarification

- Page 3 line 30: I disagree with the formulation: "(...) incorporating a stochastic weather generator (WG) that will allow for reproduction of a hydrological system response to a large variety of possible weather conditions (...)". I think you might want to say that "(...) incorporating a stochastic weather generator (WG) that will allow for a large variety of possible weather conditions that can then be provided to the hydrological model (...)". In fact, the WG itself doesn't reproduce the hydrological system's response. It produces meteorological scenarios. Then, hopefully, if the hydrological model is truly a faithful and complete representation of the hydrological system (which in many cases could be discussed), passing those weather scenarios into it will indeed provide a simulation of the basin's hydrological response.

- Page 6, line 11: Is ECOMAG really taking daily precipitation intensities as inputs? As in mm/hour? All the models I know rather use total daily precipitation. Although it is true that mm/day can be seen as an intensity (since it is a quantity over time), it seems a bit unusual to me.

- Page 8, line 29: How many months is "several"? Is it at least one full year?

- Page 9, Figure 4: On which basis did you chose to generate 1000 members from the WG while there are 50 in the ESP system? My guess is that you wanted to be sure to include a very wide variety of scenarios. While I agree that it could be interesting, (1) most operational agencies that issue meteorological forecasts from dynamic models
produce 50 ensemble members or less and (2) the number of members influences metrics such as the Continuous Ranked Probability Score (or similarly the Ranked Probability Score). See for instance Ferro et al. (2008) for a demonstration. I suggest either setting the WG to issue the same number of ensemble members as EPS or at least justifying the choice of 1000 members and discussing the impact of ensemble size on performance assessment metrics. Page 14 line 17-18: According to Murphy (1973), "Hedging is said to occur whenever a forecaster's forecast r does not correspond to his judgement p (...) ". I don't understand how you associate your results to hedging. Hedging, by definition, arise from human intervention. Since your research does not involve human forecasters, I don't think hedging is the appropriate term here. Perhaps you want to refer to a systematic over forecasting bias in the forecasting system? In my opinion, this overforecasting is to be expected if the historical database includes many years with "higher than usual" precipitations. EPS (and WG) are very much dependent on the sample of data you have.

-Page 15 line 1: What do you mean by "forecast by chance"? Please define.

-Page 15 line 10: "(...) comparing forecasts to climatology." I suspect you mean "streamflow climatology"? If so, this should be explicitly mentioned in the text everywhere applicable.

-Page 18 line 12: When you write "confidence bands", do you mean "confidence intervals"? If so, please provide the level of confidence and if not, please define what you mean by "confidence bands".

-In section 4.4: are the results for ESP or WG? Globally, the explanations in this section (page 20) are difficult to follow. In my opinion, a schematic representation of the methodology would be helpful. And since this portion is, I think, more methodological, it should be moved to section 3.3

References:
Ferro C.A.T., Richardson D.S. and Weigel A.P. (2008) On the effect of ensemble size on the discrete and continuous ranked probability scores, Meteorological Applications, 15: 19-24.

Murphy A.H. (1973) Hedging and skill scores for probability forecasts, Journal of Applied Meteorology, 12, 215-223.

3. The analysis and discussion of the results is too shallow

Section 4 of the manuscript is labelled "Results and discussion". I was therefore expecting results to be discussed (rather than simply presented) in this section. However, I find this is not the case for many figures and tables. Specifically: Table 2 (what do does values mean?), Table 3 (the "first" Table 3 on page 13) and Figure 8. I think than Table 5 could also be discussed more, since, as I mention above, the improvement over climatology is still modest. However, without any other basis of comparison (such as the current operational forecasting system), it is hard to put the results in perspective. This is also related to Figures 13 and 14, which are just presented but not discussed. Those figures show that the improvement over climatology is very modest and hence, it is difficult to appreciate the authors' contribution. Similarly, Figure 15 should be analyzed more deeply (i.e. the explanations behind the results, not simply describing the figure). In the conclusion (page 24 line 19-20), the authors implicitly mention a comparison with "the deterministic forecasts of inflow into reservoir that are used in common practice in Russia (...)", but this comparison is not explicitly shown in the manuscript.

Another thing that struck me is that the authors are not discussing the performance of their systems in terms of the relative importance of resolution and reliability. For instance on page 15 line 2, it is mentioned that the forecasts are "capable of detecting the occurrence of rare extreme events  $(\ldots)$ " This is an indication that points toward forecast reliability, but what about resolution? If the forecasts are very widely dispersed, they will likely include any events but with very low power of discrimination. This should be studied and could help to improve the discussion.
4. There are numerous spelling, orthographic and typographic errors throughout the manuscript

I will provide some specific examples in my specific comments. However, there are many such errors and I don't think it is my duty to list them exhaustively. English is not my native language either and I perfectly understand how challenging it can be to write a scientific paper in another language. From experience, I would recommend the authors to have their manuscript thoroughly proof-read by a native English speaker before resubmitting. There are also many typographic errors that should be corrected, mostly problems with parenthesis in references.

**SPECIFIC COMMENTS**

1. All figures except the first one and Figure 8 need reworking:

- Figure 2: the resolution of the right hand side figure is very poor. All the small grey pixels should be removed.

- Figure 3: I don't think this figure brings much information to the manuscript. It has no legend, and I think most readers are familiar with the requirements of a distributed, process oriented model. I suggest removing this figure.

- Figure 4: The three small figures in the center of each middle box (representing plots of time series) are much too small and of poor resolution. I suggest either modifying them to make them readable, or removing them.

- Figure 5: The x-axis is labeled 'years' while the text says "daily inflow discharge". The label of the axis should reflect what is plotted on the figure. Labeling it "years" means that you would plot yearly values, not daily.

- Figure 6: The legend is missing and the y-axis for 3 of the 4 panels need to be completed ("Simulated inflow volume, km3" rather than just "Simulated")

- Figure 7: Why is the Taylor diagram elliptical? Should it not be more spherical (a
portion of a circle)?

- Figure 9: The axes should be labeled (titles) ! Since all panels will all likely have the same axis titles, I would suggest writing axis titles only once for each: the x axis at the bottom of the figure, centered and the y axis completely on the left, also centered.

- Figures 10-11: The text on the figures (labels, ticks, etc.) is so small, it is absolutely impossible to read anything. It should be made readable, both by increasing character sizes and figure resolution. In addition, the labeling "1", "2", etc under each panel is quite unusual. I advice labeling sub-figures (a), (b), ... above each panel, as it is usually done.

- Figure 12: It is also difficult to read, although not as much as figures 10-11. The resolution of the figure could be substantially improved. Again, the labeling of the panels should be placed above, not below, each panel.

- Figures 13-14: Same thing: difficult to read. The legend is missing for Figure 14. Figure 15: Labels for panels (a, b, ...) are again misplaced. The axes ticks are very difficult to read (size and resolution).

3. Table 3 on page 13 (the "first" Table 3): the units are all missing in the first column (W, QMax, Nq and Nmax).

4. There are two tables labeled "Table 3"

5. The Taylor diagrams should be explained briefly in the methodology. At the moment, all other performance assessment tools are at least mentioned in the methodology except this one.

6. English errors and typos

(This is not an exhaustive list)

- Page 3: line 9 instead of "in (Gelfan and Motovilov 2009)", it should be "in Gelfan and Motovilov (2009)". Similarly, at line 20, remove parenthesis around 2017 in "Arnal et

**HESSD**
al. (2017)". There are many similar errors with parenthesis around references in the manuscript.

- Page 2 line 31: Change "(...) allows forecaster to provide user (...)" either to "(...) allows the forecaster to provide the user (...)" or to "(...) allows forecasters to provide users (...)"

- Page 2 line 33-34: Change "Recent studies illustrating ability of the ensemble (...)" to "Recent studies illustrating the ability of the ensemble(...)"

- Page 3, line 11: remove the "the" from "Water Problems Institute of the Russian Academy of the Sciences".

- Page 3, line 23: Change "(...) but for possible weather condition (...)" to "(...) but also for possible weather conditions (...)"

- Page 4, line 15: Replace "Also, analysis of (...)" by "Also, an analyse of (...)".

- Page 8, line 22: Replace "(...) leads to increase of the model robustness. List of the (...)" by "(...) leads to an increase of the model's robustness. A list of the (...)"

- W, Max, Nq and Nmax are sometimes in italics, sometimes not. Sometimes, the "max" in "Nmax"is in subscript ans sometimes not. Sometimes with a capital "M" and sometimes not. This needs to be uniformed according to the HESS's guidelines.

- Page 10 line 11: remove "into" in the sentence "(. . .) in which the observation fell into (. . .)"

- Page 12 line 1: Replace "Magnitude of the used metrics and error estimation has led to an assumption that the model is suitable to act as a core component of (...)" by "The magnitude of the performance assessment metrics and error estimations lead to the conclusion that the model is suitable as a core component of (...)"

- Page 12 line 12: Replace "(. . .) tested through its ability" by "(. . .) tested through their ability (. . .)"
-Page 13 line 8: Replace "For the verification purposes  $(\ldots)$ " by "For the purpose of verification  $(\ldots)$ "

- Page 13 line 8 and several other instances: replace "ensemble of the forecasted" by "ensemble forecasts"

- Page 14 line 3: I absolutely don't understand what you mean with this sentence "(...) show RMSE values around 0.55-0.65 fraction of the standard deviation of the corresponding observed characteristics." Please rephrase. In particular, which "observed characteristics" are you referring to?

- Page 15 in general (for instance line 16) and in many other places: "forecasts" should be plural. Unless you examine one single forecast for a particular time step.

- Page 17 line 10: What do you mean by "efficiency"? To me the RPSS is a measure of forecasts overall quality.

-Page 17: The mathematical variables in the text, such as Fm(j) and Om(j) should be in italics and with subscripts, as in equation (2).

- Page 20 line 25: Please correct the reference for "Report..., 2002"

HESSD

---

## Referee Comment (RC2) · Anonymous Referee #2 · 18 Aug 2017

This work describes a long-term streamflow forecasting system developed for Cheboksary reservoir inflows (Russia). The authors use the ECOMAG semi-distributed hydrological model to general ensemble forecasts via two different approaches: (1) the Ensemble Streamflow Prediction (ESP) method, and (2) weather generator (WG) based hydrologic forecasts. Forecast assessment is focused on four main metrics: (i) April-June runoff volumes, (ii) maximum inflow discharge, (iii) number of days with inflow discharge above mean observed discharge, and (iv) number of days with inflow discharge above mean maximum observed discharge. Hindcast evaluation is conducted for the period 1982-2016 (i.e., 35 water years) – using deterministic and probabilistic metrics –, and a final evaluation for 2017 is presented. The authors conclude that ESP

hindcasts are slightly better, although they provide larger confidence intervals than the WG-based technique.

Case studies like these are needed by the community in order to understand the strengths and weaknesses of existing approaches in different hydroclimatic regimes. However, I think that a fundamental flaw of this study is the lack of an overarching science question driving this effort. As a result, the manuscript reads like a technical report rather than a scientific paper, with little interpretation of the results, or even re-dundancy in the set of metrics selected and number of figures. Therefore, I think the manuscript needs a substantial revision before it can be published in HESS. I would like to encourage the authors to (1) re-think what is the gap that they intend to fill with this work – in terms of methods, information used, or forecast properties, for instance –, (2) clearly state their questions/hypotheses, and (3) re-think their current experimental design to accept or reject their hypotheses. The following comments could guide the authors to improve their work:

Major comments

1. Introduction: The authors state that "the purpose of this paper is to present the performance assessment of a long-term ensemble forecasting system of water inflow into the Cheboksary reservoir of the VKRC". I suggest re-formulating that purpose based on one or two science questions, whose answers could be found using the aforementioned system. Moreover, most of the text refers to the VKRC, with limited connection with recent literature on long-range hydrological forecasting (e.g., Schepen and Wang 2015; Mendoza et al. 2017; Beckers et al. 2016; Najafi and Moradkhani 2015; Demirel et al. 2015; Yossef et al. 2013; DeChant and Moradkhani 2014). A better link with current approaches will help readers to understand what is the contribution of this study.

2. Methods: The authors mention that the first long-term forecasts for the VKRC are dated back to the 1930s and 1940s (P2, L8). In my opinion, the authors should include

one or two benchmark methods – e.g., direct water balance methods, or index-based methods – to understand the added value of the proposed methodologies, ideally for several forecast initialization dates. Also, it is really hard for this reviewer to understand – from the information provided in the supplement section – the differences in forecast ensemble spread between ESP and WG-based technique. I think it would be helpful to see WG results contrasting boxplots or CDFs with observations for monthly precipitation amounts or temperature averages.

3. Probabilistic verification: The authors include both BS and BSS (having climatology as a reference) in Table 4, although they don't need both metrics to conclude that the WG-based approach is better for the event occurrence analyzed (similar to RPS and RPSS in table 5). Also, I strongly suggest to include some metric and/or graphic device for the assessment of forecast ensemble spread, since this is something that the authors point to without a solid quantitative basis (e.g., P16, L14). This could be done, for instance, using QQ plots (e.g., Thyer et al. 2009; Renard et al. 2010) or rank histograms (e.g., Hamill 2001; Delle Monache et al. 2006). The authors could further assess the ability of their forecasting system to distinguish between occurrence and non-occurrence by using discrimination diagrams (e.g., Clark and Slater 2006).

Minor comments

4. P6, L14: I think that the authors should provide a short description of the calibration method, since the paper should be self-contained. Also, the authors state in P6-L9 that "most of the parameters are physically meaningful". I think that statement should be re-visited, because even measurable parameters have uncertainties associated with (i) observational errors, and (ii) their applicability at spatial scales that are different to those for which physically-based equations were developed.

5. P8, L16: Please provide a reference for the Cholesky's decomposition method.

6. Verification metrics: it would be helpful to condense them in a table, including equation, possible range of values and references.

7. P10, L27: The authors state that maximum inflow discharge is well simulated by the hydrologic model, although the plot (Fig. 6) still shows considerable spread around the 1:1 line.

8. Figure 6: Instead of using "lower-left", "lower-right", etc., I suggest using panels (a), (b), (c) and (d).

9. P12, L15: "1000-year Monte Carlo generated time series". Do you mean 1000-member ensemble? Please re-word.

10. Please clarify forecast initialization dates and forecasting approach in the caption of tables and figures.

11. Figure 9: In my opinion, the results from this figure could be better communicated using time series with ensemble forecasts as boxplots, including a line with observations (e.g., Bracken et al. 2010).

12. P20, L2-25: This should be moved to the Methods section.

13. Forecast example for 2017: although this is a very interesting demonstration, I strongly encourage the authors to include verification metrics in their analyses.

References

Beckers, J. V. L. V. L., A. H. H. Weerts, E. Tijdeman, and E. Welles, 2016: ENSO-conditioned weather resampling method for seasonal ensemble streamflow prediction. Hydrol. Earth Syst. Sci., 20, 3277–3287, doi:10.5194/hess-20-3277-2016.

Bracken, C., B. Rajagopalan, and J. Prairie, 2010: A multisite seasonal ensemble streamflow forecasting technique. Water Resour. Res., 46, W03532, doi:10.1029/2009WR007965.

Clark, M. P., and A. G. Slater, 2006: Probabilistic Quantitative Precipitation Estimation in Complex Terrain. J. Hydrometeorol., 7, 3–22, doi:10.1175/JHM474.1.

[Figure]

DeChant, C. M., and H. Moradkhani, 2014: Toward a reliable prediction of seasonal forecast uncertainty: Addressing model and initial condition uncertainty with ensemble data assimilation and Sequential Bayesian Combination. J. Hydrol., 519, 2967–2977, doi:10.1016/j.jhydrol.2014.05.045.

Delle Monache, L., J. P. Hacker, Y. Zhou, X. Deng, and R. B. Stull, 2006: Probabilistic aspects of meteorological and ozone regional ensemble forecasts. J. Geophys. Res., 111, D24307, doi:10.1029/2005JD006917.

Demirel, M. C., M. J. Booij, and a. Y. Hoekstra, 2015: The skill of seasonal ensemble low-flow forecasts in the Moselle River for three different hydrological models. Hydrol. Earth Syst. Sci., 19, 275–291, doi:10.5194/hess-19-275-2015.

Hamill, T. M., 2001: Interpretation of Rank Histograms for Verifying Ensemble Forecasts. Mon. Weather Rev., 129, 550–560, doi:10.1175/1520-0493(2001)129<0550:IORHFV>2.0.CO;2.

Mendoza, P. A., A. W. Wood, E. Clark, E. Rothwell, M. P. Clark, B. Nijssen, L. D. Brekke, and J. R. Arnold, 2017: An intercomparison of approaches for improving operational seasonal streamflow forecasts. Hydrol. Earth Syst. Sci., 21, 3915–3935, doi:10.5194/hess-21-3915-2017.

Najafi, M., and H. Moradkhani, 2015: Ensemble Combination of Seasonal Streamflow Forecasts. J. Hydrol. Eng., 21, 4015043, doi:10.1061/(ASCE)HE.1943-5584.0001250.

Renard, B., D. Kavetski, G. Kuczera, M. Thyer, and S. W. Franks, 2010: Understanding predictive uncertainty in hydrologic modeling: The challenge of identifying input and structural errors. Water Resour. Res., 46, W05521, doi:10.1029/2009WR008328.

Schepen, A., and Q. J. Wang, 2015: Model averaging methods to merge operational statistical and dynamic seasonal streamflow forecasts in Australia. Water Resour. Res., 6, 1–16, doi:10.1002/2014WR016163. Thyer, M., B. Renard, D. Kavetski, G. Kuczera, S. W. Franks, and S. Srikanthan, 2009: Critical evaluation

of parameter consistency and predictive uncertainty in hydrological modeling: A case study using Bayesian total error analysis. Water Resour. Res., 45, W00B14, doi:10.1029/2008WR006825.

Yossef, N. C., H. Winsemius, A. Weerts, R. Van Beek, and M. F. P. Bierkens, 2013: Skill of a global seasonal streamflow forecasting system, relative roles of initial conditions and meteorological forcing. Water Resour. Res., 49, 4687–4699, doi:10.1002/wrcr.20350.
* * *

---

## Referee Comment (RC3) · Anonymous Referee #3 · 20 Aug 2017

This paper presents a case study on developing the long-term ensemble forecast of water inflow into the Cherkasy reservoir using ECOMAG model with two forecasting schemes: ESP and weather generator. Four inflow characteristics related to inflow volume, maximum value, and timing were evaluated using a set of probabilistic and deterministic metrics. The modeling and analysis was carried out within the framework of the Phata Rhei Research Initiative of the International Association of Hydrologic Sciences. However, my major concern is that this paper does not provide any insight or broader implications to hydrologic forecasting beyond the Cheboksary reservoir. My comments are given as follows:

1. In the introduction section, the authors need to re-formulate their research objective based on the current state of literature. The current objective "present the performance assessment of a long-term ensemble forecasting system of water inflow into the Chebokary reservoir of the VKRC" seems too restrictive to a specific area.

2. This study compared ESP and weather generator forecasting schemes. These two methods are classic approaches for inflow forecasts and have been tested in many regions. The ESP approach is based on the ensemble of historical observed weather data. The weather generator approach generates synthetic weather data based on stochastic models. These two approaches also generated a different number of ensemble members: 50 versus 1000. This is like comparing apple and orange. The discussion and description of these two methods are too shallow. The authors need to clarify the motivations and implications of comparing these two forecasting schemes, with in-depth analysis and comments on these two methods.

3. When evaluating probabilistic forecast, the authors used Brier skill score to compare the two forecast schemes with climatology of the inflow. The weather forecast forcing constructed using ESP is actually climatology of the weather variables. To compare these two forecasting schemes, I think it would be helpful to use ESP as a reference forecast relative to the WG-based forecast.

---

## Author Comment (AC1) · 16 Sep 2017

**Authors' responses to the comments of anonymous Reviewer 1**

We would like to thank Reviewer 1 for the important and constructive criticisms and suggestions made to our manuscript. We substantially revise the manuscript in accordance with the suggestions. The main revisions are the following:

(1) we fully re-write Introduction to point out the existing gaps in the area of interest and thereby to clarify motivation and objective of the study;

(2) we include subsections containing description of the current operational forecast of inflow to the Cheboksary reservoir and comparison of the developed model-based forecast with the operational one;

(3) we substantially revise the Results and Discussion section to emphasize our contribution, and (4) priori to resubmission we'll have a language check done by a native speaker..

Below, we respond to the Reviewer's comments in a point-by-point manner.

**_1. The original contribution is not clear/significant enough_**

*To sum up my first major comment: I think the authors should clarify and emphasize their original contribution, which is not clear for me at the moment. One way to do so, besides phrasing it more clearly, is to include a comparison with at least one of the 3 operational forecasting systems described on page 2.*

We revise Introduction to highlight the motivation of the study and our original contribution. We include new subsection 3.3.1 describing the current operational forecasting method and new subsection 4.3.2 describing comparison of the developed model-based forecast with the operational one.

*In fact, many of the results presented in the manuscript do not appear to me as a clear improvement over climatology. For instance, in Table 5, the skill scores obtained by the WG-based forecasts are all below 0.5. While I do agree that this represents an improvement over climatology, it is not a large one.*

We agree that the RPSS estimates are quite low and do not demonstrate clear improvement over climatology. In the revised Discussion section we consider this result and express our point of view on the possible ways for the forecast improvement. Not in order to justify the rather weak result, we'd like to note here that the values of RPSS<0.5 are not infrequently presented in well-cited publications related to the ensemble streamflow forecast verification (see, for instance, Greel et al., 2016 (Fig. 8c); Yuan et al., 2012 (Fig. 4); Franz et al., 2003 (Fig. 8))

Greuell W., Franssen W. H. P., Biemans Hester, Hutjes Ronald W. A. (2016) Seasonal streamflow forecasts for Europe – I. Hindcast verification with pseudo- and real observations. HESSD, doi:10.5194/hess-2016-603, 2016

Yuan X., Wood E.F., Roundy J.K. and Ming Pan (2012) CFSv2-based seasonal hydroclimatic forecasts over the conterminous United States. J. Climate, 26, 4828-4847

Franz J K Hartmann H C Sorooshian S and Bales R 2003 Verification of National Weather Service Ensemble Streamflow Predictions for water supply forecasting in the Colorado River Basin J. Hydrometeorology 4 1105-1118.

*I don't have any problem with this (a slight improvement over climatology), but it would probably be much more convincing to see (also) the improvement relative to at least one of the current operational methods mentioned on page 3.*

The improvement over the operational forecast is demonstrated in subsection 4.3.2.

*Figures 13 and 14 also support my comment: the forecasts presented on Figure 14 appear only slightly different from the climatology presented on Figure 13. This is especially true for the forecasts issued on March 1st (Figure 14 left) compared to Figure 13.*

Indeed, the hydrograph predicted for the spring season of 2017 is close to the climatic one, but this proximity of the hydrographs is largely occasional. In most of 35 years, the predicted hydrographs were significantly different from the climatic one. As an illustration of this statement, Fig. 1R shows difference between forecasted and climatic hydrographs for some years of the verification period.

[Figure]

**Figure 1R Forecast of daily inflow into the Cheboksary reservoir during March – June in selected years compared to the climatic mean inflow**

**2. Some methodological/conceptual elements need clarification**

*Page 3 line 30: I disagree with the formulation: "(. . .) incorporating a stochastic weather generator (WG) that will allow for reproduction of a hydrological system response to a large variety of possible weather conditions (. . .)". I think you might want to say that "(. . .) incorporating a stochastic weather generator (WG) that will allow for a large variety of possible weather conditions that can then be provided to the hydrological model (. . .)".*

We agree with the Reviewer and revise the fragment in accordance with the suggestion.

*Page 6, line 11: Is ECOMAG really taking daily precipitation intensities as inputs? As in mm/hour? All the models I know rather use total daily precipitation. Although it is true that mm/day can be seen as an intensity (since it is a quantity over time), it seems a bit unusual to me.*

Precipitation intensity (L/T) as well as other flows (evaporation, infiltration, streamflow, etc.) is contained in the ECOMAG governing equations (see Motovilov et al., 1999). Since these equations are numerically integrated under 1-day time step, than ECOMAG really takes daily precipitation intensity in mm/day.

Motovilov, Yu., Gottschalk, L., Engeland, K., and Belokurov, A.: ECOMAG – regional model of hydrological cycle. Application to the NOPEX region. Department of Geophysics, University of Oslo, Institute Report Series no. 105. 1999.

*Page 8, line 29: How many months is "several"? Is it at least one full year?*

We have edited the respective paragraph in order to clarify this

(1) Spin-up ECOMAG-based simulations ("warm start") using meteorological observation data prior to the forecast issue date in order to calculate the initial watershed hydrological state (soil, snow and channel water contents, groundwater level, soil freezing depth, etc.) that initializes the forecast. The simulations start from the end of the previous freshet, i.e. 8-9 months before the forecast issue date

*Page 9, Figure 4: On which basis did you chose to generate 1000 members from the WG while there are 50 in the ESP system? I suggest either setting the WG to issue the same number of ensemble members as EPS or at least justifying the choice of 1000 members and discussing the impact of ensemble size on performance assessment metrics.*

We add several fragments relating to this issue into the revised manuscript. First of all, we include additional literature review in the Introduction (Buizza and Palmer, 1998; Richardson 2001; Müller et al. 2005; Weigel et al., 2007; Ferro et al. 2008; Najafi et al. 2012) and conclude that the forecast skill is improved "as the ensemble size increases, wherein degree of improvement depends on the verification measure used". In the Result section we highlight that the ranked probability skill score (RPSS) is strongly dependent on ensemble size and negatively biased. Then we add estimations of RPSS bias into the corresponding table (Table 5 in the first version of the manuscript) and show that the bias of the ESP-based forecast is two orders of magnitude larger than that of the WF-based forecast. In the revised manuscript, dependence of the RPSS bias on sample size is analyzed and the illustrating figure is added (see below as Fig. 2R). One can see from this Fig. 2R that under the used 35-member ensemble (i.e. the ESP-based ensemble) the bias can reach tens of percent depending on the RPSS estimate. Under the used 1000-member ensemble, the bias is close to zero.

[Figure]

Fig. 2R. Negative bias of the RPSS-estimate in dependence on an ensemble size

Buizza, R., and T. N. Palmer, 1998: Impact of ensemble size on ensemble prediction. Mon. Wea. Rev., 126, 2503–2518.

Ferro, C. A. T., Richardson, D. S., and Weigel, A. P.: On the effect of ensemble size on the discrete and continuous ranked probability scores, Meteorol. Appl., 15, 19–24, 2008.

Müller WA, Appenzeller C, Doblas-Reyes FJ, Liniger MA. 2005. A debiased ranked probability skill score to evaluate probabilistic ensemble forecasts with small ensemble sizes. Journal of Climate 18: 1513–1523.

Najafi, M. R., Moradkhani, H., and Piechota, T. C.: Climate signal weighting methods vs. Climate Forecast System Reanalysis, J. Hydrol., 442–443, 105–116, 2012.

Richardson, D. S.: Measures of skill and value of ensemble prediction systems, their interrelationship and the effect of ensemble size, Q. J. Roy. Meteorol. Soc., 127, 2473–2489, 2001

Weigel AP, Liniger MA, Appenzeller C. 2007. The discrete Brier and ranked probability skill scores. Monthly Weather Review 135: 118–124.

*Page 14 line 17-18: According to Murphy (1973), "Hedging is said to occur whenever a forecaster's forecast r does not correspond to his judgement p (...) ". I don't understand how you associate your results to hedging. Hedging, by definition, arise from human intervention. Since your research does not involve human forecasters, I don't think hedging is the appropriate term here. Perhaps you want to refer to a systematic over forecasting bias in the forecasting system? In my opinion, this overforecasting is to be expected if the historical database includes many years with "higher than usual" precipitations. EPS (and WG) are very much dependent on the sample of data you have.*

We revise the following sentence according to the reviewer's comment:
> The hindcasts of Qmax show perfect detection estimates for both methodologies, but, as the frequency Bias is very high, this might be an outcome of overprediction, so with the high values of False Alarm Ratio and Hansen-Kuipers score.

*Page 15 line 1: What do you mean by "forecast by chance"? Please define*

We clarify the respective fragment as follows
> However, the forecast accuracy with Heidke Skill Score of more than 60% is significantly better tan the accuracy of random chance

*Page 15 line 10: "(. . .) comparing forecasts to climatology." I suspect you mean "streamflow climatology"? If so, this should be explicitly mentioned in the text everywhere applicable.*

Changed in accordance to the suggestion.

*Page 18 line 12: When you write "confidence bands", do you mean "confidence intervals"? If so, please provide the level of confidence and if not, please define what you mean by "confidence bands"*

Indeed, confidence bands are closely related to confidence intervals and in some cases they are synonyms (see, for instance, Owen, 1995). Thus we use "confidence interval" term in the revised text and explicitly define the level of confidence in Figs. 10-11.
> Owen, A.B. (1995). Nonparametric likelihood confidence bands for a distribution function. *J. American Stat. Association.* 90(430): 516–521

*In section 4.4: are the results for ESP or WG? Globally, the explanations in this section (page 20) are difficult to follow. In my opinion, a schematic representation of the methodology would be helpful. And since this portion is, I think, more methodological, it should be moved to section 3.3*

Section 4.4 is removed from the revised manuscript because the forecastability issues turned to be out of the main framework of the study after the revisions

**3. The analysis and discussion of the results is too shallow**

*Section 4 of the manuscript is labelled "Results and discussion". I was therefore expecting results to be discussed (rather than simply presented) in this section. However, I find this is not the case for many figures and tables. Specifically:*

*Table 2 (what do does values mean?),*

In the revised version of the manuscript, it is clearly pointed out that Table 2 demonstrates meaning and values of the WG parameters

*Table 3 (the "first" Table 3 on page 13) and Figure 8.*

In the revised supplement section, all used verification measures are condensed in a Table 1S, including equation, possible range of values and references.

*I think than Table 5 could also be discussed more, since, as I mention above, the improvement over climatology is still modest. However, without any other basis of comparison (such as the current operational forecasting system), it is hard to put the results in perspective. This is also related to Figures 13 and 14, which are just presented but not discussed. Those figures show that the improvement over climatology is very modest and hence, it is difficult to appreciate the authors' contribution. Similarly, Figure 15 should be analyzed more deeply (i.e. the explanations behind the results, not simply describing the figure).*

The discussion section is substantially enhanced in accordance with the Reviewer's recommendation

*In the conclusion (page 24 line 19-20), the authors implicitly mention a comparison with "the deterministic forecasts of inflow into reservoir that are used in common practice in Russia (. . .)", but this comparison is not explicitly shown in the manuscript.*

We include new subsection 3.3.1 describing the current operational forecasting method and new subsection 4.3.2 describing comparison of the developed model-based forecast with the operational one

*Another thing that struck me is that the authors are not discussing the performance of their systems in terms of the relative importance of resolution and reliability. For instance on page 15 line 2, it is mentioned that the forecasts are "capable of detecting the occurrence of rare extreme events (. . .)" This is an indication that points toward forecast reliability, but what about resolution? If the forecasts are very widely dispersed, they will likely include any events but with very low power of discrimination. This should be studied and could help to improve the discussion.*

Discrimination and reliability diagrams are presented and analyzed in the Results and Discussion section of the revised manuscript.

4. There are numerous spelling, orthographic and typographic errors throughout the manuscript

*4.1 All figures except the first one and Figure 8 need reworking:*

*Figure 2: the resolution of the right hand side figure is very poor. All the small grey pixels should be removed.*

Figure 2 is removed from the revised manuscript

*Figure 3: I don't think this figure brings much information to the manuscript. It has no legend, and I think most readers are familiar with the requirements of a distributed, process oriented model. I suggest removing this figure.*

Figure 3 is removed in accordance with the Reviewer's suggestion

*Figure 4: The three small figures in the center of each middle box (representing plots of time series) are much too small and of poor resolution. I suggest either modifying them to make them readable, or removing them*

Removed in accordance with the Reviewer's suggestion

*Figure 5: The x-axis is labeled 'years' while the text says "daily inflow discharge". The label of the axis should reflect what is plotted on the figure. Labeling it "years" means that you would plot yearly values, not daily.*

Corrected according to the reviewer's recommendation.

*Figure 6: The legend is missing and the y-axis for 3 of the 4 panels need to be completed ("Simulated inflow volume, km^3" rather than just "Simulated")*

Corrected according to the reviewer's recommendation.

*Figure 7: Why is the Taylor diagram elliptical? Should it not be more spherical (a portion of a circle)?*

Corrected according to the reviewer's recommendation.

*Figure 9: The axes should be labeled (titles) ! Since all panels will all likely have the same axis titles, I would suggest writing axis titles only once for each: the x axis at the bottom of the figure, centered and the y axis completely on the left, also centered.*

Corrected according to the reviewer's recommendation.

*Figures 10-11: The text on the figures (labels, ticks, etc.) is so small, it is absolutely impossible to read anything. It should be made readable, both by increasing character sizes and figure resolution. In addition, the labeling "1", "2", etc under each panel is quite unusual. I advice labeling sub-figures (a), (b), . . . above each panel, as it is usually*

*done.*

Corrected according to the reviewer's recommendation.

*Figure 12: It is also difficult to read, although not as much as figures 10-11. The resolution of the figure could be substantially improved. Again, the labeling of the panels should be placed above, not below, each panel.*

Figure 12 is removed from the revised manuscript

*Figures 13-14: Same thing: difficult to read. The legend is missing for Figure 14.*

Corrected according to the reviewer's recommendation.

*Figure 15: Labels for panels (a, b, . . .) are again misplaced. The axes ticks are very difficult to read (size and resolution).*

Corrected according to the reviewer's recommendation.

*4.2. Table 3 on page 13 (the "first" Table 3): the units are all missing in the first column (W, QMax, Nq and Nmax).*

Corrected according to the reviewer's recommendation.

*4.3 There are two tables labeled "Table 3"*

Corrected according to the reviewer's recommendation.

*4.4 The Taylor diagrams should be explained briefly in the methodology. At the moment, all other performance assessment tools are at least mentioned in the methodology except this one.*

Corrected according to the reviewer's recommendation. The following fragment is added to the text:

To illustrate forecast performance we used the Taylor diagram (Taylor, 2001) as it combines three forecast characteristics in one chart, namely the forecast standard deviation, RMSE and the correlation coefficient between the observations and the forecasted values. The values of all characteristics are normalized by dividing the RMSE and the standard deviations of the forecasts by the standard deviation of the observations. This normalization provides a vivid demonstration of the forecast efficiency expressed by RMSE fraction of the observed standard deviation (grey circular lines in Fig. 6). As long as the forecast RMSE is lower than the standard deviation of the observations, the forecast can be considered efficient against climatology.

5. English errors and typos

*Page 3: line 9 instead of "in (Gelfan and Motovilov 2009)", it should be "in Gelfan and Motovilov (2009)". Similarly, at line 20, remove parenthesis around 2017 in "Arnal et al. (2017)". There are many similar errors with parenthesis around references in the manuscript.*

Corrected according to the reviewer's recommendation.

*Page 2 line 31: Change "(. . .) allows forecaster to provide user (. . .)" either to "(. . .) allows the forecaster to provide the user (. . .)" or to "(. . .) allows forecasters to provide users (. . .)"*

Corrected according to the reviewer's recommendation.

*Page 2 line 33-34: Change "Recent studies illustrating ability of the ensemble (.. .)" to "Recent studies illustrating the ability of the ensemble(. . .)".*

Corrected according to the reviewer's recommendation.

*Page 3, line 11: remove the "the" from "Water Problems Institute of the Russian Academy of the Sciences*

Corrected according to the reviewer's recommendation.

*Page 3, line 23: Change "(. . .) but for possible weather condition (. . .)" to "(. . .) but also for possible weather conditions (. . .)".*

Corrected according to the reviewer's recommendation.

*Page 4, line 15: Replace "Also, analysis of (. . .)" by "Also, an analyse of (. . .)".*

Corrected according to the reviewer's recommendation.

*Page 8, line 22: Replace "(. . .) leads to increase of the model robustness. List of the (. . .)" by "(. . .) leads to an increase of the model's robustness.*

Corrected according to the reviewer's recommendation.

*A list of the (. . .)" W, Max, Nq and Nmax are sometimes in italics, sometimes not. Sometimes, the "max" in "Nmax"is in subscript ans sometimes not. Sometimes with a capital "M" and sometimes not. This needs to be uniformed according to the HESS's guidelines.*

Corrected according to the reviewer's recommendation.

*Page 10 line 11: remove "into" in the sentence "(. . .) in which the observation fell into (. . .)"*

Corrected according to the reviewer's recommendation.

*Page 12 line 1: Replace "Magnitude of the used metrics and error estimation has led to an assumption that the model is suitable to act as a core component of (. . .)" by "The magnitude of the performance assessment metrics and error estimations lead to the conclusion that the model is suitable as a core component of (. . .)".*

Corrected according to the reviewer's recommendation.

*Page 12 line 12: Replace "(. . .) tested through its ability" by "(. . .) tested through their ability (. . .)*

Corrected according to the reviewer's recommendation.

---

## Author Comment (AC2) · 16 Sep 2017

**Authors' responses to the comments of anonymous Reviewer 2**

We would like to thank Reviewer 2 for the important and constructive criticisms and suggestions made to our manuscript. We substantially revise the manuscript in accordance with the suggestions. The main revisions are the following: (1) we fully re-write Introduction to point out the existing gaps in the area of interest and thereby to clarify motivation and objective of the study; (2) we include subsections containing description of the current operational forecast of inflow to the Cheboksary reservoir and comparison of the developed model-based forecast with the operational one; (3) we substantially revise the Result and Discussion section to stress our contribution, and (4) priori to resubmission we'll have a language check done by a native English speaker..

Below, we respond to the Reviewer's comments in a point-by-point manner.

*1. Introduction: The authors state that "the purpose of this paper is to present the performance assessment of a long-term ensemble forecasting system of water inflow into the Cheboksary reservoir of the VKRC". I suggest re-formulating that purpose based on one or two science questions, whose answers could be found using the aforementioned system.*

We revise Introduction to highlight motivation of the study. The corresponding fragments of the revised text are below:

> …utilizing the process-oriented hydrological models results in strengthening of physical adequacy of the forecast and, potentially, in improving forecast accuracy in comparison with the operational practice. However, this potential is rarely studied; to our knowledge the only example is the comprehensive experiment presented by Mendoza et al. (2017) and comparing the ESP model-based forecasts with the operational data-driven forecasts for a multi-year historical period. Our paper partly bridges this gap. We present development and verification of the ESP-based forecasts of water inflow into the Cheboksary reservoir of the VKRC and compare them against the operational forecasts for 35 years.
>
> …The observed weather scenarios that are used within the ESP framework do not encompass all of the possible weather conditions for the forecast period. … Hence the ensemble size is limited to the number of the historical years, statistical problems can appear stemming from large sample errors. For instance, Buizza and Palmer (1998) demonstrate improvement of the weather forecast skill as the ensemble size increases, wherein degree of improvement depends on the verification measure used. Particularly, the ranked probability skill score is strongly dependent on ensemble size and negatively biased (see also Müller et al. 2005, Weigel et al., 2007). Different aspects of the ensemble size effect on statistical properties of the ensemble weather forecast and verification scores are studied by Richardson (2001), Ferro et al. (2008), Najafi et al. (2012).The problem, can be solved by incorporating a stochastic weather generator (WG) into the ESP procedure… In this paper, we compare the ESP-based forecast with the WG-based forecast and assess possible advantage of the latter approach in forecasting rare hydrological events in the study basin and estimating verification measures.
>
> > Thus, the motivation of this study is to answer two questions: (1) Does the model-based ESP technique allow one to improve reliability and skill of the operational forecast of spring inflow into the Cheboksary reservoir? (2) Does the enlarged ensemble size lead to any appreciable advantage when using the WG-simulated ensemble compared to the ESP-based ensemble?

*2. …Most of the text refers to the VKRC, with limited connection with recent literature on long-range hydrological forecasting (e.g., Schepen and Wang 2015; Mendoza et al. 2017; Beckers et al. 2016; Najafi and Moradkhani 2015; Demirel et al. 2015; Yossef et al. 2013; DeChant and Moradkhani 2014). A better link with current approaches will help readers to understand what is the contribution of this study.*

In the revised manuscript, review of the recent literature is added, in particular using the listed publications

*3. Methods. The authors mention that the first long-term forecasts for the VKRC are dated back to the 1930s and 1940s (P2, L8). In my opinion, the authors should include one or two benchmark methods – e.g., direct water balance methods, or index-based methods – to understand the added value of the proposed methodologies, ideally for several forecast initialization dates*

We include new subsection 3.3.1 describing the current operational forecasting method and new subsection 4.3.2 describing comparison of the developed model-based forecast with the operational one

*4. ...it is really hard for this reviewer to understand – from the information provided in the supplement section – the differences in forecast ensemble spread between ESP and WG-based technique. I think it would be helpful to see WG results contrasting boxplots or CDFs with observations for monthly precipitation amounts or temperature averages*

In accordance with the Reviewer's suggestion, we add the corresponding boxplots to the supplement section (Fig. 8S).

*5. The authors include both BS and BSS (having climatology as a reference) in Table 4, although they don't need both metrics to conclude that the WG-based approach is better for the event occurrence analyzed (similar to RPS and RPSS in table 5). Also, I strongly suggest to include some metric and/or graphic device for the assessment of forecast ensemble spread, since this is something that the authors point to without a solid quantitative basis (e.g., P16, L14). This could be done, for instance, using QQ plots (e.g., Thyer et al. 2009; Renard et al. 2010) or rank histograms (e.g., Hamill 2001; Delle Monache et al. 2006). The authors could further assess the ability of their forecasting system to distinguish between occurrence and non-occurrence by using discrimination diagrams (e.g., Clark and Slater 2006).*

Discrimination diagrams and Q-Q plots are presented and analyzed in the Result and Discussion section of the revised manuscript.

Minor comments

*1. P6, L14: I think that the authors should provide a short description of the calibration method, since the paper should be self-contained. Also, the authors state in P6-L9 that "most of the parameters are physically meaningful". I think that statement should be re-visited, because even measurable parameters have uncertainties associated with (i) observational errors, and (ii) their applicability at spatial scales that are different to those for which physically-based equations were developed.*

The calibration method is described briefly in the sub-section 3.1 of the revised manuscript
The ECOMAG calibration procedure is described in detail by Gelfan et al. (2015). Here, we emphasize two issues concerning the procedure. First, values of a few key-parameters pre-assigned from literature of from available measurements are considered as the initial approximation of the optimal values and the latter are quested within the closest neighborhood of the initial, pre-assigned values. Second, in the process of calibration, the ratios between the initial values of the distributed parameter relating to different soils, landscapes and vegetation are conserved.

*2. P8, L16: Please provide a reference for the Cholesky's decomposition method*

Reference is included
Press W.H., Teukolsky S.A., Vetterling W.T., Flannery B.P. Numerical recipes: the art of scientific computing: 2.9 Cholesky decomposition, 3rd Edition, Cambridge University Press, 2007.

*6. Verification metrics: it would be helpful to condense them in a table, including equation, possible range of values and references.*

Verification metrics are summarized in new Table 1S added into the supplement section

*7. P10, L27: The authors state that maximum inflow discharge is well simulated by the hydrologic model, although the plot (Fig. 6) still shows considerable spread around the 1:1 line.*

The authors apologize for the fact that this particular panel was constructed erroneously. We revised the figure and added a one-standard-deviation-wide confidence band to the 1:1 line so one can see that there are only a few points outside this band.

*8. Figure 6: Instead of using "lower-left", "lower-right", etc., I suggest using panels (a), (b), (c) and (d)*

Corrected according to the reviewer's recommendation.

*9. P12, L15: "1000-year Monte Carlo generated time series". Do you mean 1000- member ensemble? Please re-word.*

Corrected according to the reviewer's recommendation.

*10. Please clarify forecast initialization dates and forecasting approach in the caption of tables and figures.*

Corrected according to the reviewer's recommendation.

*11. Figure 9: In my opinion, the results from this figure could be better communicated using time series with ensemble forecasts as boxplots, including a line with observations (e.g., Bracken et al. 2010)*

We'd prefer keeping this figure as is. In our opinion, CDFs better demonstrate performance of probabilistic forecast than boxplots

*12. P20, L2-25: This should be moved to the Methods section.*

Section 4.4 is removed from the revised manuscript because the forecastability issues turned to be out of the main framework of the study after the revisions

*13. Forecast example for 2017: although this is a very interesting demonstration, I strongly encourage the authors to include verification metrics in their analyses.*

Additional verification metrics are included

---

## Author Comment (AC3) · 16 Sep 2017

**Authors' responses to the comments of anonymous Reviewer 3**

We would like to thank Reviewer 3 for the important and constructive criticisms and suggestions made to our manuscript. We substantially revise the manuscript in accordance with the suggestions. The main revisions are the following:

(1) we fully re-write Introduction to point out the existing gaps in the area of interest and thereby to clarify motivation and objective of the study;

(2) we include subsections containing description of the current operational forecast of inflow to the Cheboksary reservoir and comparison of the developed model-based forecast with the operational one;

(3) we substantially revise the Results and Discussion section to emphasize our contribution, and

(4) priori to resubmission we'll have a language check done by a native speaker..

Below, we respond to the Reviewer's comments in a point-by-point manner.

*1. In the introduction section, the authors need to re-formulate their research objective based on the current state of literature. The current objective "present the performance assessment of a long-term ensemble forecasting system of water inflow into the Chebokary reservoir of the VKRC " seems too restrictive to a specific area*

We revise Introduction to highlight the motivation of the study and our original contribution. The corresponding fragments of the revised introductory section are below:

> …utilizing the process-oriented hydrological models results in strengthening of physical adequacy of the forecast and, potentially, in improving forecast accuracy in comparison with the operational practice. However, this potential is rarely studied; to our knowledge the only example is the comprehensive experiment presented by Mendoza et al. (2017) and comparing the ESP model-based forecasts with the operational data-driven forecasts for a multi-year historical period. Our paper partly bridges this gap. We present development and verification of the ESP-based forecasts of water inflow into the Cheboksary reservoir of the VKRC and compare them against the operational forecasts for 35 years.

> …The observed weather scenarios that are used within the ESP framework do not encompass all of the possible weather conditions for the forecast period. … Hence the ensemble size is limited to the number of the historical years, statistical problems can appear stemming from large sample errors. For instance, Buizza and Palmer (1998) demonstrate improvement of the weather forecast skill as the ensemble size increases, wherein degree of improvement depends on the verification measure used. Particularly, the ranked probability skill score is strongly dependent on ensemble size and negatively biased (see also Müller et al. 2005,  Weigel et al., 2007). Different aspects of the ensemble size effect on statistical properties of the ensemble weather forecast and verification scores are studied by Richardson (2001), Ferro et al. (2008), Najafi et al. (2012).The problem, can be solved by incorporating a stochastic weather generator (WG) into the ESP procedure… In this paper, we compare the ESP-based forecast with the WG-based forecast and assess possible advantage of the latter approach in forecasting rare hydrological events in the study basin and estimating verification measures.
>
> Thus, the motivation of this study is to answer two questions:  (1) Does the model-based ESP technique allow one to improve reliability and skill of the operational forecast of spring inflow into the Cheboksary reservoir? (2) Does the enlarged ensemble size lead to any appreciable advantage when using the WG-simulated ensemble compared to the ESP-based ensemble?

*2. This study compared ESP and weather generator forecasting schemes. These two methods are classic approaches for inflow forecasts and have been tested in many regions...*

Indeed, the WG-based scheme is classic approach for inflow forecasts and has been tested in many regions. However this is the case for short-term forecasts but not for the long-term ones. Particularly, there are only a few attempts to use synthetic, stochastically generated time series of weather variables instead of the historical data within the ESP framework. Hanes et al. (1977) were probably the first who used Monte-Carlo simulated sequences of daily precipitation to drive the conceptual US Geological Survey hydrological model and provide ensemble seasonal forecast of snowmelt runoff volume. A physically-based distributed hydrological model was used in combination with a weather generator to create a long-term probabilistic forecast of spring runoff of rivers in Central Russia by Kuchment and Gelfan (2007) and Gelfan et al. (2015). Caraway et al. (2014) incorporated a stochastic weather generator into the ESP to make a probabilistic seasonal climate forecasts and applied the modified methodology to the San Juan River snowmelt dominated basin. Beckers et al. (2016) uses ENSO-conditioned weather generator to compensate for the reduction of ensemble size in the post-processing ensemble forecast scheme presented for the Columbia River basin.

In the listed papers, there are no attempts to compare the ESP-based forecast with the WG-based forecast. Our study bridges this gap.

*…The ESP approach is based on the ensemble of historical observed weather data. The weather generator approach generates synthetic weather data based on stochastic models. These two approaches also generated a different number of ensemble members: 50 versus 1000. This is like comparing apple and orange…*

In our opinion, comparison of the ESP-based and the WG-based approaches makes sense because allows us to highlight the problem of limited ensemble size when evaluating the first approach.

We include additional literature review in the Introduction (Buizza and Palmer, 1998; Richardson 2001; Müller et al. 2005; Weigel et al., 2007; Ferro et al. 2008; Najafi et al. 2012) and conclude that the forecast skill is improved "as the ensemble size increases, wherein degree of improvement depends on the verification measure used". In the Result section we argue that the ranked probability skill score (RPSS) depends on the ensemble size and negatively biased. Then we add estimations of the RPSS bias into the revised Table 5 and show that the bias of the ESP-based forecast is two orders of magnitude larger than that of the WF-based forecast. In the revised manuscript, dependence of the RPSS bias on sample size is analyzed and the illustrating figure is added (see below as Fig. 1R). One can see from this figure that under the used 35-member ensemble (i.e. the ESP-based ensemble) the bias can reach tens of percent depending on the RPSS estimate. Under the used 1000-member ensemble, the bias is close to zero.

[Figure]

Fig. 1R. Negative bias of the RPSS-estimate in dependence on an ensemble size

*The discussion and description of these two methods are too shallow. The authors need to clarify the motivations and implications of comparing these two forecasting schemes, with in-depth analysis and comments on these two methods.*

In the revised manuscript, the Discussion section is substantially enhanced in accordance with the Reviewer's recommendation.

*3. When evaluating probabilistic forecast, the authors used Brier skill score to compare the two forecast schemes with climatology of the inflow. The weather forecast forcing constructed using ESP is actually climatology of the weather variables. To compare these two forecasting schemes, I think it would be helpful to use ESP as a reference forecast relative to the WG-based forecast.*

Thank you for the comment. Indeed, it is very interesting to use ESP as a reference forecast and we include the corresponding results into the revised manuscript.

---

## Author Response (AR1)

**Authors' responses to the comments of anonymous Reviewer 1**

We would like to thank Reviewer 1 for the important and constructive criticisms and the suggestions made. We have substantially revised the manuscript in accordance with the suggestions. The main revisions are in the following:

(1) we have completely re-written Introduction to highlight the existing gaps in the area of interest and thereby to clarify motivation and objective of the study;

(2) we have included subsections containing description of the current operational forecast of inflow to the Cheboksary reservoir and comparison of the developed model-based forecast with the operational one;

(3) we have substantially revised the Results and Discussion section to emphasize our contribution, and

(4) we have considerably revised English.

Below, we are responding to the Reviewer's comments, point-by-point.

**1. The original contribution is not clear/significant enough**

*To sum up my first major comment: I think the authors should clarify and emphasize their original contribution, which is not clear for me at the moment. One way to do so, besides phrasing it more clearly, is to include a comparison with at least one of the 3 operational forecasting systems described on page 2.*

We have revised Introduction to highlight the motivation of the study and our original contribution. We have included new subsection 3.1 ("Operational data-driven forecast of spring inflow into the Cheboksary reservoir: a current practice") describing the current operational forecasting method and new subsection 4.3.1 ("Ensemble (model-based) and operational (data-driven) deterministic forecasts") describing comparison of the developed model-based forecast with the operational one.

*In fact, many of the results presented in the manuscript do not appear to me as a clear improvement over climatology. For instance, in Table 5, the skill scores obtained by the WG-based forecasts are all below 0.5. While I do agree that this represents an improvement over climatology, it is not a large one.*

We agree that the RPSS estimates are quite low and do not demonstrate clear improvement over climatology. In the revised Discussion section we consider this result and express our point of view on the possible ways for the forecast improvement. Without an attempt to characterise this result as a strong one (since it is of course not very strong), we'd like to note here that the values of RPSS<0.5 are not infrequent in the well-cited publications related to the ensemble streamflow forecast verification (see, for instance, Greel et al., 2016 (Fig. 8c); Yuan et al., 2012 (Fig. 4); Franz et al., 2003 (Fig. 8))

Greuell W., Franssen W. H. P., Biemans Hester, Hutjes Ronald W. A. (2016) Seasonal streamflow forecasts for Europe – I. Hindcast verification with pseudo- and real observations. HESSD, doi:10.5194/hess-2016-603, 2016

Yuan X., Wood E.F., Roundy J.K. and Ming Pan (2012) CFSv2-based seasonal hydroclimatic forecasts over the conterminous United States. J. Climate, 26, 4828-4847

Franz J K Hartmann H C Sorooshian S and Bales R 2003 Verification of National Weather Service Ensemble Streamflow Predictions for water supply forecasting in the Colorado River Basin J. Hydrometeorology 4 1105-1118.

*I don't have any problem with this (a slight improvement over climatology), but it would probably be much more convincing to see (also) the improvement relative to at least one of the current operational methods mentioned on page 3.*

The improvement over the operational forecast is demonstrated in subsection 4.3.1 ("Ensemble (model-based) and operational (data-driven) deterministic forecasts").

*Figures 13 and 14 also support my comment: the forecasts presented on Figure 14 appear only slightly different from the climatology presented on Figure 13. This is especially true for the forecasts issued on March 1$^{st}$ (Figure 14 left) compared to Figure 13.*

Indeed, the hydrograph predicted for the spring season of 2017 is close to the climatic one, but this proximity of the hydrographs is largely incidental. In most of 35 years, the predicted hydrographs were significantly different from the climatic one. As an illustration of this statement, Fig. 1R shows difference between forecasted and climatic hydrographs for some years of the verification period.

[Figure]

**Figure 1R Forecast of daily inflow into the Cheboksary reservoir during March – June in selected years compared to the climatic mean inflow**

**2. Some methodological/conceptual elements need clarification**

*Page 3 line 30: I disagree with the formulation: "(. . .) incorporating a stochastic weather generator (WG) that will allow for reproduction of a hydrological system response to a large variety of possible weather conditions (. . .)". I think you might want to say that "(. . .) incorporating a stochastic weather generator (WG) that will allow for a large variety of possible weather conditions that can then be provided to the hydrological model (. . .)".*

We agree with the Reviewer and this fragment has been revised.

*Page 6, line 11: Is ECOMAG really taking daily precipitation intensities as inputs? As in mm/hour? All the models I know rather use total daily precipitation. Although it is true that mm/day can be seen as an intensity (since it is a quantity over time), it seems a bit unusual to me.*

Precipitation intensity (L/T) as well as other flows (evaporation, infiltration, streamflow, etc.) is contained in the ECOMAG governing equations (see Motovilov et al., 1999). Since these equations are numerically integrated under 1-day time step, ECOMAG actually takes daily precipitation intensity in mm/day.

Motovilov, Yu., Gottschalk, L., Engeland, K., and Belokurov, A.: ECOMAG – regional model of hydrological cycle. Application to the NOPEX region. Department of Geophysics, University of Oslo, Institute Report Series no. 105. 1999.

*Page 8, line 29: How many months is "several"? Is it at least one full year?*

Agreed - it was not clear. We have revised the respective paragraph in order to clarify this

(1) Spin-up ECOMAG-based simulations ("warm start") using meteorological observations data prior to the forecast issue date (March, 31) in order to calculate the initial watershed hydrological state (soil, snow and channel water contents, groundwater level, soil freezing depth, etc.) that initializes the forecast. The simulations start from the end of the previous freshet, i.e. 8-9 months before the forecast issue date;

*Page 9, Figure 4: On which basis did you chose to generate 1000 members from the WG while there are 50 in the ESP system? I suggest either setting the WG to issue the same number of ensemble members as EPS or at least justifying the choice of 1000 members and discussing the impact of ensemble size on performance assessment metrics.*

Good point. We have revised the text adding several fragments related to this issue. First of all, we have included additional literature review in the Introduction (Buizza and Palmer, 1998; Richardson 2001; Müller et al. 2005; Weigel et al., 2007; Ferro et al. 2008; Najafi et al. 2012) and concluded that the forecast skill is improved "as the ensemble size increases, wherein degree of improvement depends on the verification measure used". In the Result section we highlight that the ranked probability skill score (RPSS) is strongly dependent on ensemble size and negatively biased. In the revised manuscript, dependence of the RPSS bias on sample size is analyzed and the illustrating figure is added (see below as Fig. 2R). One can see from this Fig. 12 that under the used 51-member ensemble (i.e. the ESP-based ensemble) the bias can reach tens of percent depending on the RPSS estimate. Under the used 1000-member ensemble, the bias is close to zero.

[Figure]

**Figure 2R: Negative bias of the RPSS-estimate in dependence on the ensemble size and the RPSS value**

Buizza, R., and T. N. Palmer, 1998: Impact of ensemble size on ensemble prediction. Mon. Wea. Rev., 126, 2503–2518.

Ferro, C. A. T., Richardson, D. S., and Weigel, A. P.: On the effect of ensemble size on the discrete and continuous ranked probability scores, Meteorol. Appl., 15, 19–24, 2008.

Müller WA, Appenzeller C, Doblas-Reyes FJ, Liniger MA. 2005. A debiased ranked probability skill score to evaluate probabilistic ensemble forecasts with small ensemble sizes. Journal of Climate 18: 1513–1523.

Najafi, M. R., Moradkhani, H., and Piechota, T. C.: Climate signal weighting methods vs. Climate Forecast System Reanalysis, J. Hydrol., 442–443, 105–116, 2012.

Richardson, D. S.: Measures of skill and value of ensemble prediction systems, their interrelationship and the effect of ensemble size, Q. J. Roy. Meteorol. Soc., 127, 2473–2489, 2001

Weigel AP, Liniger MA, Appenzeller C. 2007. The discrete Brier and ranked probability skill scores. Monthly Weather Review 135: 118–124.

*Page 14 line 17-18: According to Murphy (1973), "Hedging is said to occur whenever a forecaster's forecast r does not correspond to his judgement p (...) ". I don't understand how you associate your results to hedging. Hedging, by definition, arise from human intervention. Since your research does not involve human forecasters, I don't think hedging is the appropriate term here. Perhaps you want to refer to a systematic over forecasting bias in the forecasting system? In my opinion, this overforecasting is to be expected if the historical database includes many years with "higher than usual" precipitations. EPS (and WG) are very much dependent on the sample of data you have.*

Good point. We have revised the following sentence according to the reviewer's comment:

> The forecasts show good detection estimates (even perfect for $Q_{max}$) for both model-based methodologies. However, as the frequency Bias is high, this might be the result of overprediction, so with the high values of False Alarm Ratio and Hansen-Kuipers score.

*Page 15 line 1: What do you mean by "forecast by chance"? Please define*

We have clarified the respective fragment as follows

> For W and Qmax, the forecast accuracy with the Heidke Skill Score (HSS) of around 60% is better than the accuracy of random chance

*Page 15 line 10: "(. . .) comparing forecasts to climatology." I suspect you mean "streamflow climatology"? If so, this should be explicitly mentioned in the text everywhere applicable.*

Agrred. Changed in accordance to this suggestion.

*Page 18 line 12: When you write "confidence bands", do you mean "confidence intervals"? If so, please provide the level of confidence and if not, please define what you mean by "confidence bands"*

Indeed, confidence bands are closely related to confidence intervals and in some cases they are synonyms (see, for instance, Owen, 1995). Thus we use "confidence interval" term in the revised text and explicitly define the level of confidence in Figs. 10-11.

> Owen, A.B. (1995). Nonparametric likelihood confidence bands for a distribution function. *J. American Stat. Association*. 90(430): 516–521

*In section 4.4: are the results for ESP or WG? Globally, the explanations in this section (page 20) are difficult to follow. In my opinion, a schematic representation of the*

*methodology would be helpful. And since this portion is, I think, more methodological, it should be moved to section 3.3*

Agreed. Section 4.4 is removed from the revised manuscript because the predictability issues turned to be out of the main framework of this study after the revisions.

*3. The analysis and discussion of the results is too shallow*

*Section 4 of the manuscript is labelled "Results and discussion". I was therefore expecting results to be discussed (rather than simply presented) in this section. However, I find this is not the case for many figures and tables. Specifically:*

*Table 2 (what do does values mean?),*

In the revised version of the manuscript, it is clearly pointed out that this Table (replaced in the Supplementary Material) presents the meaning and the values of the WG parameters

*Table 3 (the "first" Table 3 on page 13) and Figure 8.*

In the revised supplement section, all the verification measures used are condensed in a Table 1S, including equation, possible range of values and references.

*I think than Table 5 could also be discussed more, since, as I mention above, the improvement over climatology is still modest. However, without any other basis of comparison (such as the current operational forecasting system), it is hard to put the results in perspective. This is also related to Figures 13 and 14, which are just presented but not discussed. Those figures show that the improvement over climatology is very modest and hence, it is difficult to appreciate the authors' contribution. Similarly, Figure 15 should be analyzed more deeply (i.e. the explanations behind the results, not simply describing the figure).*

The discussion section is now substantially reworked and enhanced in accordance with the Reviewer's recommendation. Particularly, former Fig. 15 (Fig. 7 in the revised paper) is analyzed, as well as new Fig. 8 is added to compare the ensemble forecasts with the current operational forecasts.

*In the conclusion (page 24 line 19-20), the authors implicitly mention a comparison with "the deterministic forecasts of inflow into reservoir that are used in common practice in Russia (. . .)", but this comparison is not explicitly shown in the manuscript.*

Agreed. We have included the new subsection 3.1 ("Operational data-driven forecast of spring inflow into the Cheboksary reservoir: a current practice") describing the current operational forecasting method and new subsection 4.3.1 ("Ensemble (model-based) and operational (data-driven) deterministic forecasts") describing comparison of the developed model-based forecast with the operational one.

*Another thing that struck me is that the authors are not discussing the performance of their systems in terms of the relative importance of resolution and reliability. For instance on page 15 line 2, it is mentioned that the forecasts are "capable of detecting the occurrence of*

*rare extreme events (. . .)" This is an indication that points toward forecast reliability, but what about resolution? If the forecasts are very widely dispersed, they will likely include any events but with very low power of discrimination. This should be studied and could help to improve the discussion.*

Agreed. Discrimination and reliability diagrams are added into the Supplimentary material and are discussed in the Results and Discussion section of the revised manuscript.

4. There are numerous spelling, orthographic and typographic errors throughout the manuscript

*4.1 All figures except the first one and Figure 8 need reworking:*

*Figure 2: the resolution of the right hand side figure is very poor. All the small grey pixels should be removed.*

Figure 2 has been removed from the revised manuscript

*Figure 3: I don't think this figure brings much information to the manuscript. It has no legend, and I think most readers are familiar with the requirements of a distributed, process oriented model. I suggest removing this figure.*

Figure 3 has been removed in accordance with the Reviewer's suggestion

*Figure 4: The three small figures in the center of each middle box (representing plots of time series) are much too small and of poor resolution. I suggest either modifying them to make them readable, or removing them*

Removed in accordance with the Reviewer's suggestion

*Figure 5: The x-axis is labeled 'years' while the text says "daily inflow discharge". The label of the axis should reflect what is plotted on the figure. Labeling it "years" means that you would plot yearly values, not daily.*

(Fig. 2 in the revised paper) Corrected according to the reviewer's recommendation.

*Figure 6: The legend is missing and the y-axis for 3 of the 4 panels need to be completed ("Simulated inflow volume, km^3" rather than just "Simulated")*

(Fig. 3 in the revised paper) Corrected according to the reviewer's recommendation.

*Figure 7: Why is the Taylor diagram elliptical? Should it not be more spherical (a portion of a circle)?*

(Fig. 4 in the revised paper) Corrected according to the reviewer's recommendation.

*Figure 9: The axes should be labeled (titles)! Since all panels will all likely have the same axis titles, I would suggest writing axis titles only once for each: the x axis at the bottom of the figure, centered and the y axis completely on the left, also centered.*

Corrected according to the reviewer's recommendation.

*Figures 10-11: The text on the figures (labels, ticks, etc.) is so small, it is absolutely impossible to read anything. It should be made readable, both by increasing character sizes and figure resolution. In addition, the labeling "1", "2", etc under each panel is quite unusual. I advice labeling sub-figures (a), (b), . . . above each panel, as it is usually done.*

(Fig. 11 in the revised paper) Corrected according to the reviewer's recommendation.

*Figure 12: It is also difficult to read, although not as much as figures 10-11. The resolution of the figure could be substantially improved. Again, the labeling of the panels should be placed above, not below, each panel.*

Figure 12 has been removed from the revised manuscript

*Figures 13-14: Same thing: difficult to read. The legend is missing for Figure 14.*

Figure 13 has been removed from the revised manuscript. Fig. 14 (Fig 6 in the revised version) has been corrected according to the reviewer's recommendation.

*Figure 15: Labels for panels (a, b, . . .) are again misplaced. The axes ticks are very difficult to read (size and resolution).*

(Fig. 7 in the revised paper) Corrected according to the reviewer's recommendation.

*4.2. Table 3 on page 13 (the "first" Table 3): the units are all missing in the first column (W, QMax, Nq and Nmax).*

(Table 1 in the revised paper). Corrected according to the reviewer's recommendation.

*4.3 There are two tables labeled "Table 3"*

Corrected.

*4.4 The Taylor diagrams should be explained briefly in the methodology. At the moment, all other performance assessment tools are at least mentioned in the methodology except this one.*

Corrected according to the reviewer's recommendation. The following fragment is added to the text:

To compare the ESP-based and the WG-based forecasts, we present them in the form of the Taylor diagram (Fig. 5; Taylor, 2001), which combines three forecast characteristics in one chart, namely, the forecast standard deviation, RMSE and the correlation coefficient between the observed and the forecasted values of the inflow characteristics. The values of all characteristics are normalized by dividing the RMSE by the standard deviation of the observations. This normalization provides a demonstration of the forecast efficiency expressed in fractions of the observed standard deviation. As long as the forecast RMSE is less than the standard deviation of the observations, the forecast can be considered efficient against climatology.

5. English errors and typos

*Page 3: line 9 instead of "in (Gelfan and Motovilov 2009)", it should be "in Gelfan and Motovilov (2009)". Similarly, at line 20, remove parenthesis around 2017 in "Arnal et al. (2017)". There are many similar errors with parenthesis around references in the manuscript.*

Corrected according to the reviewer's recommendation.

*Page 2 line 31: Change "(. . .) allows forecaster to provide user (. . .)" either to "(. . .) allows the forecaster to provide the user (. . .)" or to "(. . .) allows forecasters to provide users (. . .)"*

The fragment has been removed

*Page 2 line 33-34: Change "Recent studies illustrating ability of the ensemble (.. .)" to "Recent studies illustrating the ability of the ensemble(. . .)".*

The fragment has been removed

*Page 3, line 11: remove the "the" from "Water Problems Institute of the Russian Academy of the Sciences*

Corrected according to the reviewer's recommendation.

*Page 3, line 23: Change "(. . .) but for possible weather condition (. . .)" to "(. . .) but also for possible weather conditions (. . .)".*

The fragment has been removed

*Page 4, line 15: Replace "Also, analysis of (. . .)" by "Also, an analyse of (. . .)".*

The fragment has been removed

*Page 8, line 22: Replace "(. . .) leads to increase of the model robustness. List of the (. . .)" by "(. . .) leads to an increase of the model's robustness.*

The fragment has been removed.

*A list of the (. . .)" W, Max, Nq and Nmax are sometimes in italics, sometimes not. Sometimes, the "max" in "Nmax"is in subscript ans sometimes not. Sometimes with a capital "M" and sometimes not. This needs to be uniformed according to the HESS's guidelines.*

Corrected according to the reviewer's recommendation.

*Page 10 line 11: remove "into" in the sentence "(. . .) in which the observation fell into (. . .)"*

Corrected according to the reviewer's recommendation.

*Page 12 line 1: Replace "Magnitude of the used metrics and error estimation has led to an assumption that the model is suitable to act as a core component of (. . .)" by "The magnitude of the performance assessment metrics and error estimations lead to the conclusion that the model is suitable as a core component of (. . .)".*

The fragment has been removed

*Page 12 line 12: Replace "(. . .) tested through its ability" by "(. . .) tested through their ability (. . .)*

Corrected according to the reviewer's recommendation.

**Authors' responses to the comments of anonymous Reviewer 2**

We would like to thank Reviewer 2 for the important and constructive criticisms and suggestions made to our manuscript. We have substantially revised the manuscript in accordance with the suggestions. The main revisions are in the following:

(1) we have completely re-written Introduction to point out the existing gaps in the area of interest and thereby to clarify motivation and objective of the study;

(2) we have included subsections containing description of the current operational forecast of inflow to the Cheboksary reservoir and comparison of the developed model-based forecast with the operational one;

(3) we have substantially revised the Results and Discussion section to emphasize our contribution, and

(4) we have considerably revised English.

Below, we respond to the Reviewer's comments in a point-by-point manner.

*1. Introduction: The authors state that "the purpose of this paper is to present the performance assessment of a long-term ensemble forecasting system of water inflow into the Cheboksary reservoir of the VKRC". I suggest re-formulating that purpose based on one or two science questions, whose answers could be found using the aforementioned system.*

Agreed. We have revised Introduction to highlight motivation of the study. The corresponding fragments of the revised text follow:

> …utilizing the process-oriented hydrological models results in increasing the physical adequacy of forecasts and, potentially, in improving forecast accuracy in comparison with the methods currently used in operational practice. However, such quantitative comparisons is not a common place; to our knowledge the only example is the comprehensive experiment presented by Mendoza et al. (2017) and comparing the ESP model-based forecasts with the operational data-driven forecasts for a multi-year historical period.

> …The observed weather scenarios that are used within the ESP framework do not encompass all of the possible weather conditions for the forecast period. It is desirable to account not only for the observed weather, but for possible weather condition that might lead to freshet events of rare occurrence. Assessing the magnitude of such an event might be crucial for decision making. Moreover, hence the ensemble size is limited to the number of the historical years, statistical problems can appear stemming from large sample errors. For instance, Buizza and Palmer (1998) demonstrate improvement of the weather forecast skill as the ensemble size increases, wherein degree of improvement depends on the verification measure used. Particularly, the ranked probability skill score is strongly dependent on ensemble size and negatively biased (see also Müller et al. 2005, Weigel et al., 2007). Different aspects of the ensemble size effect on statistical properties of the ensemble weather forecast and verification scores are studied by Richardson (2001), Ferro et al. (2008), Najafi et al. (2012). The problem can be solved by incorporating synthetic, stochastically generated time series of weather variables instead of the historical data used within the ESP framework.

> …The studies and examples mentioned above serve as the background and the main motivation for this study. The objective of this study is to contribute to the EPS-related studies, with the focus on the comparative analysis between the data-driven techniques used in operational forecasts, and the ensemble forecasts of sreamflow, using two different weather scenarios: a) based on the historical data, and b) employing the WG-based forecasts. The case study is the Cheboksary reservoir of the VKRC cascade for which the operational forecasts are available since 1982.
> Thus, this study is an attempt to to answer the following two research questions: (1) Does the model-based ensemble methodology allow one to improve reliability and skill of the operational forecast of spring inflow into the Cheboksary reservoir, and to what extent? (2) Does the enlarged ensemble size lead to any noticeable advantage when using the WG-simulated ensemble compared to the ESP-based ensemble?

*2. ...Most of the text refers to the VKRC, with limited connection with recent literature on long-range hydrological forecasting (e.g., Schepen and Wang 2015; Mendoza et al. 2017; Beckers et al. 2016; Najafi and Moradkhani 2015; Demirel et al. 2015; Yossef et al. 2013; DeChant and Moradkhani 2014). A better link with current approaches will help readers to understand what is the contribution of this study.*

In the revised manuscript, review of the recent literature is added, in particular including the listed publications

*3. Methods. The authors mention that the first long-term forecasts for the VKRC are dated back to the 1930s and 1940s (P2, L8). In my opinion, the authors should include one or two benchmark methods – e.g., direct water balance methods, or index-based methods – to understand the added value of the proposed methodologies, ideally for several forecast initialization dates*

We have included new subsection 3.1 ("Operational data-driven forecast of spring inflow into the Cheboksary reservoir: a current practice") describing the current operational forecasting method and new subsection 4.3.1 ("Ensemble (model-based) and operational (data-driven) deterministic forecasts") describing comparison of the developed model-based forecast with the operational one.

*4. ...it is really hard for this reviewer to understand – from the information provided in the supplement section – the differences in forecast ensemble spread between ESP and WG-based technique. I think it would be helpful to see WG results contrasting boxplots or CDFs with observations for monthly precipitation amounts or temperature averages*

Agreed. In accordance with the Reviewer's suggestion, we add the corresponding boxplots to the supplement section (Fig. 8S).

*5. The authors include both BS and BSS (having climatology as a reference) in Table 4, although they don't need both metrics to conclude that the WG-based approach is better for the event occurrence analyzed (similar to RPS and RPSS in table 5). Also, I strongly suggest to include some metric and/or graphic device for the assessment of forecast ensemble spread, since this is something that the authors point to without a solid quantitative basis (e.g., P16, L14). This could be done, for instance, using QQ plots (e.g., Thyer et al. 2009; Renard et al. 2010) or rank histograms (e.g., Hamill 2001; Delle Monache et al. 2006). The authors could further assess the ability of their forecasting system to distinguish between occurrence and non-occurrence by using discrimination diagrams (e.g., Clark and Slater 2006).*

Agreed. Discrimination diagrams and Q-Q plots are presented and analyzed in the Result and Discussion section of the revised manuscript.

Minor comments

*1. P6, L14: I think that the authors should provide a short description of the calibration method, since the paper should be self-contained. Also, the authors state in P6-L9 that "most of the parameters are physically meaningful". I think that statement should be re-visited, because even measurable parameters have uncertainties associated with (i) observational errors, and (ii) their applicability at spatial scales that are different to those for which physically-based equations were developed.*

The calibration method is described briefly in the sub-section 3.1 of the revised manuscript

The ECOMAG calibration procedure is described in detail by Gelfan et al. (2015). Here, we emphasize the two issues concerning this procedure. First, the values of several key parameters pre-assigned from literature or from the available measurements are considered as the initial approximations of the optimal values and the latter are sought within the neighborhood of the initial, pre-assigned values. Second, during the calibration process, the ratios between the initial values of the distributed parameter corresponding to different soils, landscapes and vegetation are preserved. The Nash and Sutcliffe (1970) efficiency criterion NSE is adopted to represent the goodness of fit of the simulated and measured variables.

*2. P8, L16: Please provide a reference for the Cholesky's decomposition method*

Reference is included

Press W.H., Teukolsky S.A., Vetterling W.T., Flannery B.P. Numerical recipes: the art of scientific computing: 2.9 Cholesky decomposition, 3rd Edition, Cambridge University Press, 2007.

*6. Verification metrics: it would be helpful to condense them in a table, including equation, possible range of values and references.*

Verification metrics are summarized in new Table 1S added into the supplement section

*7. P10, L27: The authors state that maximum inflow discharge is well simulated by the hydrologic model, although the plot (Fig. 6) still shows considerable spread around the 1:1 line.*

Indeed. The authors apologize for the fact that this particular panel contained errors. We have revised the figure and added a one-standard-deviation-wide confidence band to the 1:1 line so one can see that there are only a few points outside this band.

*8. Figure 6: Instead of using "lower-left", "lower-right", etc., I suggest using panels (a), (b), (c) and (d)*

Corrected according to the reviewer's recommendation (Fig. 3 in the revised version).

*9. P12, L15: "1000-year Monte Carlo generated time series". Do you mean 1000- member ensemble? Please re-word.*

Corrected according to the reviewer's recommendation.

*10. Please clarify forecast initialization dates and forecasting approach in the caption of tables and figures.*

Corrected.

*11. Figure 9: In my opinion, the results from this figure could be better communicated using time series with ensemble forecasts as boxplots, including a line with observations (e.g., Bracken et al. 2010)*

We appreciate this suggestion, and discussed it, however, still would prefer to keep this figure as is. In our opinion, CDFs better demonstrate performance of probabilistic forecast than boxplots.

*12. P20, L2-25: This should be moved to the Methods section.*

Section 4.4 is removed from the revised manuscript because the predictability issues turned to be out of the main framework of the study after the revisions

*13. Forecast example for 2017: although this is a very interesting demonstration, I strongly encourage the authors to include verification metrics in their analyses.*

We have included comparison of the ESP-based and operational forecasts of water inflow into the Cheboksary reservoir for the period of 01/04/2017-30/06/2017 (Fig. 8)

**Authors' responses to the comments of anonymous Reviewer 3**

We would like to thank Reviewer 3 for the important and constructive criticisms and suggestions made to our manuscript. We have substantially revised the manuscript in accordance with the suggestions. The main revisions are in the following:

(1) we have fully re-written Introduction to point out the existing gaps in the area of interest and thereby to clarify motivation and objective of the study;

(2) we have included subsections containing description of the current operational forecast of inflow to the Cheboksary reservoir and comparison of the developed model-based forecast with the operational one;

(3) we have substantially revised the Results and Discussion section to emphasize our contribution, and

(4) we have revised Enlgish.

Below, we respond to the Reviewer's comments in a point-by-point manner.

*1. In the introduction section, the authors need to re-formulate their research objective based on the current state of literature. The current objective "present the performance assessment of a long-term ensemble forecasting system of water inflow into the Chebokary reservoir of the VKRC " seems too restrictive to a specific area*

We have revised Introduction to highlight the motivation of the study and our original contribution. The corresponding fragments of the revised introductory section are below:

> …utilizing the process-oriented hydrological models results in increasing the physical adequacy of forecasts and, potentially, in improving forecast accuracy in comparison with the methods currently used in operational practice. However, such quantitative comparisons is not a common place; to our knowledge the only example is the comprehensive experiment presented by Mendoza et al. (2017) and comparing the ESP model-based forecasts with the operational data-driven forecasts for a multi-year historical period.

> …The observed weather scenarios that are used within the ESP framework do not encompass all of the possible weather conditions for the forecast period. It is desirable to account not only for the observed weather, but for possible weather condition that might lead to freshet events of rare occurrence. Assessing the magnitude of such an event might be crucial for decision making. Moreover, hence the ensemble size is limited to the number of the historical years, statistical problems can appear stemming from large sample errors. For instance, Buizza and Palmer (1998) demonstrate improvement of the weather forecast skill as the ensemble size increases, wherein degree of improvement depends on the verification measure used. Particularly, the ranked probability skill score is strongly dependent on ensemble size and negatively biased (see also Müller et al. 2005, Weigel et al., 2007). Different aspects of the ensemble size effect on statistical properties of the ensemble weather forecast and verification scores are studied by Richardson (2001), Ferro et al. (2008), Najafi et al. (2012). The problem can be solved by incorporating synthetic, stochastically generated time series of weather variables instead of the historical data used within the ESP framework.

> …The studies and examples mentioned above serve as the background and the main motivation for this study. The objective of this study is to contribute to the EPS-related studies, with the focus on the comparative analysis between the data-driven techniques used in operational forecasts, and the ensemble forecasts of sreamflow, using two different weather scenarios: a) based on the historical data, and b) employing the WG-based forecasts. The case study is the Cheboksary reservoir of the VKRC cascade for which the operational forecasts are available since 1982.
> Thus, this study is an attempt to to answer the following two research questions: (1) Does the model-based ensemble methodology allow one to improve reliability and skill of the operational forecast of spring inflow into the Cheboksary reservoir, and to what extent? (2) Does the enlarged

ensemble size lead to any noticeable advantage when using the WG-simulated ensemble compared to the ESP-based ensemble?

*2. This study compared ESP and weather generator forecasting schemes. These two methods are classic approaches for inflow forecasts and have been tested in many regions...*

Indeed, the WG-based scheme is a classic approach for inflow forecasts and has been tested in many regions. However it was mainly done for the short-term forecasts but not for the long-term ones. Particularly, there are not too many attempts to use stochastic weather generator (WG) within framework of long-term ensemble forecasting. Hanes et al. (1977) were probably the first who used Monte-Carlo simulated sequences of daily precipitation to drive the conceptual US Geological Survey hydrological model and provide ensemble seasonal forecast of snowmelt runoff volume. A physically-based distributed hydrological model was used in combination with a weather generator to create a long-term probabilistic forecast of spring runoff of rivers in Central Russia in Kuchment and Gelfan (2007), Gelfan et al. (2015). Caraway et al. (2014) incorporated a stochastic weather generator into the ESP to make a probabilistic seasonal climate forecasts and applied the modified methodology to the San Juan River snowmelt dominated basin. Beckers et al. (2016) used ENSO-conditioned weather generator to compensate for the reduction of ensemble size in the post-processing ensemble forecast scheme presented for the Columbia River basin

In the listed papers, there have been no attempts to compare the ESP-based forecast with the WG-based forecast. Our study bridges this gap.

*...The ESP approach is based on the ensemble of historical observed weather data. The weather generator approach generates synthetic weather data based on stochastic models. These two approaches also generated a different number of ensemble members: 50 versus 1000. This is like comparing apple and orange...*

In our opinion, comparison of the ESP-based and the WG-based approaches makes sense because allows us to highlight the problem of limited ensemble size when evaluating the first approach.

We have included additional literature review in the Introduction (Buizza and Palmer, 1998; Richardson 2001; Müller et al. 2005; Weigel et al., 2007; Ferro et al. 2008; Najafi et al. 2012) and conclude that the forecast skill is improved "as the ensemble size increases, wherein degree of improvement depends on the verification measure used". In the Result section we highlight that the ranked probability skill score (RPSS) is strongly dependent on ensemble size and negatively biased. In the revised manuscript, dependence of the RPSS bias on sample size is analyzed and the illustrating figure is added (see below as Fig. 1R). One can see from this Fig. 12 that under the used 51-member ensemble (i.e. the ESP-based ensemble) the bias can reach tens of percent depending on the RPSS estimate. Under the used 1000-member ensemble, the bias is close to zero.

[Figure]

**Figure 1R: Negative bias of the RPSS-estimate in dependence on the ensemble size and the RPSS value**

*The discussion and description of these two methods are too shallow. The authors need to clarify the motivations and implications of comparing these two forecasting schemes, with in-depth analysis and comments on these two methods.*

In the revised manuscript, the Discussion section is substantially enhanced in accordance with the Reviewer's recommendation.

*3. When evaluating probabilistic forecast, the authors used Brier skill score to compare the two forecast schemes with climatology of the inflow. The weather forecast forcing constructed using ESP is actually climatology of the weather variables. To compare these two forecasting schemes, I think it would be helpful to use ESP as a reference forecast relative to the WG-based forecast.*

Thank you for the comment. Indeed, it is interesting to use ESP as a reference forecast and we have included the corresponding results in the revised manuscript.

---

## Referee Report (RR1)

**Review of "Long-term ensemble forecast of snowmelt inflow into the Cheboksary reservoir under the two different weather scenarios" by Gelfan et al. (hess-2017-389)**

The manuscript by Gelfan et al. is clear and well written. The authors have conducted a well thought out study that presents a long-term ensemble methodology applied to water inflows into the Cheboksary reservoir (Volga river, central part of European Russia). They used a semi-distributed hydrological model to calculate an ensemble of hydrographs using two different sets of weather ensembles: ESP (Ensemble Streamflow Prediction) based forecast constructed from observed weather data, and WG (Weather generator) based forecast that simulates synthetic data using a multi-site weather generator on the basis of monte-carlo simulations.

The importance of the study area is emphasised in the introduction and the literature review is well presented, highlighting the gaps in the research field and explaining the scientific contribution of this paper in comparison to previous studies. Research questions are clear and the method section is well structured as it is presented schematically in form of subsections.

However, I believe more room needs to be given for discussions on section 4, in particular in a few subsections, as in my opinion some of the results are presented yet not discussed. More specifically:

-   Subsection 4.2: Results of parameter estimation and model testing are described and referred to the supplementary materials, however I did not find an actual "discussion" on the findings.
-   Subsection 4.3.2: Figure 6 is presented and shortly described yet I expected a brief discussion on the behaviour of daily forecast ensembles in comparison to the observed inflow data before seeing the boxplots on figure 7.

On section 5 I recommend to summarize the main findings of the investigation in only two main points, in order to match the research questions, thus points 1 and 2 could be merged into one. In addition, instead of discussing/presenting current and future work specific to the authors I would prefer to read your opinion on future advancements on the research field in general on the basis of your findings. Hence I suggest rephrasing this last paragraph.

Overall, I believe this research is very much in line with the topic of this special issue called "Sub-seasonal to seasonal hydrological forecasting", therefore I trust that after addressing the comments within this review and some revisions it can be considered for potential publication in HESS.

Best regards,

**Specific comments:**

P3.L33: Replace "sreamflow" with "streamflow"

P9.L28: The notation "Figure x" is used here while in the rest of the manuscript the notation adopted was "Fig. x". Please choose one formatting and adapt in order to be consistent.

P13.L18: Replace "respecrively" with "respectively"

P20.L5-11: I would prefer to see this as part of section 4.

P23.L6-15: I would prefer to see this as part of section 4, from "A two sided…"

P24.L12: Remove the parenthesis in "(Weigel et al., 2007)"

Figure 1: Please improve figure resolution, whereas words are a bit hard to read. Also check the legend, I am not sure what the orange circle with black dots represents since it is not mentioned anywhere in the text. If they are not necessary please remove them from the map.

Figure 3: Please re-arrange order in the caption so as to present panels in alphabetical order, I believe this would make the reading easier. I also suggest removing the black line since it is not necessary and brings confusion on the plot.

Figure 5: I find the legend confusing since black triangles and black circles supposedly represent ESP and WG based forecasts in general while at the same time represent ESP and WG based forecasts for "number of days with inflow discharge above maximum ($N_{qMax}$)". Instead of representing "W", "Qmax", "Nq" and "Nqmax" with coloured squares, which are not observed after in the Taylor diagram, I suggest to use a different type of polygon or symbol representing "W", "Qmax", "Nq" and "Nqmax", maybe empty instead, and to colour them on the basis of ESP or WG based forecasts. I also suggest making polygons or symbols larger, since they look rather small compare to the entire graph. In this way, one colour defines whether the forecast is ESP of WG based and the polygon/symbol indicates type of forecast.

Figure 7: Please re-arrange order in the caption so as to present panels in alphabetical order, I believe this would make the reading easier. Also describe in the caption what is the black continuous line representing.

Figure 9: Colours described in the caption do not match the observed in the figure. Please correct.

Figure 10: I am not sure if is a resolution problem or type of file format, however colours are kind of washed out in this figure. It would be better to see it as the other figures.

Table 1: I would prefer to see the measurement units on the first column in between parenthesis.

Table 2: Same as table 1.

References:

P28.L25: Replace "Sum-mary" with "Summary"

---

## Author Response (AR2)

**Authors' responses to the comments of anonymous Reviewer 1**

We would like to thank Reviewer 1 for the constructive criticisms and the suggestions made. We have revised the manuscript in accordance with the suggestions. Below, we are responding to the Reviewer's comments, point-by-point.

*1. Please check the use of the words "significant" and "significantly" throughout the manuscript, since readers may associate them with "statistically significant" – even if you are not meaning that.*

Corrected according to the reviewer's recommendation.

*2. P6, L7-8: I think that an explanation on how you obtain deterministic forecasts from ensemble forecasts – although provided later in the manuscript – should go here*

Corrected according to the reviewer's recommendation.

*3. P10, L8-11: The authors might want to consider putting these results into a table, for both calibration and validation periods.*

We would prefer keeping these results as a text, not a table

*4. P12, L10-11: The think the ensemble sizes (51 and 1000) should be specified in the Method section*

Corrected according to the reviewer's recommendation.

*5. Conclusions: I suggest the authors connecting their future work with existing literature on data assimilation for hydrological forecasting (e.g., Clark et al. 2006; McMillan et al. 2013; Dechant and Moradkhani 2011; DeChant and Moradkhani 2014; Huang et al. 2016) and weather forecast post-processing – also referred to as pre-processing (e.g., Hamill et al. 2004, 2008; Fraley et al. 2010; Schmeits and Kok 2010; Verkade et al. 2013; Crochemore et al. 2016)*

Several suggested references have been added according to the reviewer's recommendation.

**Authors' responses to the comments of anonymous Reviewer 2**

We would like to thank Reviewer 2 for the constructive criticisms and the suggestions made. We have revised the manuscript in accordance with the suggestions. Below, we are responding to the Reviewer's comments, point-by-point.

*1. Subsection 4.2: Results of parameter estimation and model testing are described and referred to the supplementary materials, however I did not find an actual "discussion" on the findings.*

Discussion is enhanced in accordance with the Reviewer's recommendation. The following fragment is added.

Figs 1S and 8S demonstrate the ability of the developed weather generator to reproduce annual and monthly mean values of air temperature, precipitation and humidity deficit. Fig. 8S demonstrates good correspondence between the distributions of the observed and modelled precipitation, as well as Fig. 2S where a good match between the observed and the modelled coefficient of variation can be seen. Despite some bias, the model errors do not appear to be systematic. The ability of the generator to preserve the spatial structure of the weather variables was examined by evaluating the spatial correlation curves (Fig. 7s) for temperature and precipitation, which demonstrate close match for both daily temperature and precipitation.

*2. Subsection 4.3.2: Figure 6 is presented and shortly described yet I expected a brief discussion on the behaviour of daily forecast ensembles in comparison to the observed inflow data before seeing the boxplots on figure 7*

Brief discussion is added in accordance with the Reviewer's recommendation.

Fig. 6 shows the outcome of the anomalous weather conditions that led to earlier increase of the inflow in mid-March (see panel (a)), which was not captured by the mean ensemble hydrograph of the forecast issued on March 1st. However, several scenarios of the ensemble show the behavior of inflow similar to the observed one. The forecast issued on March 27th showed the ongoing increase in inflow discharge, however the colder weather conditions led to inflow stabilization, not captured by the forecast. One can see visible improvement of the mean ensemble hydrograph issued on March 27th (Fig. 6b) comparing with the one issued on March 1st (Fig. 6a).

*3. On section 5 I recommend to summarize the main findings of the investigation in only two main points, in order to match the research questions, thus points 1 and 2 could be merged into one. In addition, instead of discussing/presenting current and future work specific to the authors I would prefer to read your opinion on future advancements on the research field in general on the basis of your findings. Hence I suggest rephrasing this last paragraph.*

Points 1 and 2 are merged into one, to fit the research questions, and the last paragraph is corrected according to the reviewer's recommendation.

*P3.L33: Replace "sreamflow" with "streamflow"*
Corrected according to the reviewer's recommendation.

*P9.L28: The notation "Figure x" is used here while in the rest of the manuscript the notation adopted was "Fig. x". Please choose one formatting and adapt in order to be consistent.*
Corrected according to the reviewer's recommendation.

*P13.L18: Replace "respecrively" with "respectively"*
Corrected according to the reviewer's recommendation.

*P20.L5-11: I would prefer to see this as part of section 4.*
We appreciate this comment, and the reason for it (to have all "methods" in one place, which is, formally speaking, a usual arrangement). However, after discussion, we came to a conclusion (or rather a suggestion) that for a reader's benefit, it would be better to keep this particular piece (Eq 4) in section 4.3.3. From our point of view, it would be easier for a reader to appreciate Fig 9 which immediately follows Eq 4 which explains how CDF is formed.

*P23.L6-15: I would prefer to see this as part of section 4, from "A two sided..."*
We appreciate this comment, but would like to present the same arguments, as just have been given for the previous comment, and suggest to keep Equ 6-9 . in section 4.3.4.

*P24.L12: Remove the parenthesis in "(Weigel et al., 2007)"*
Corrected according to the reviewer's recommendation.

*Figure 1: Please improve figure resolution, whereas words are a bit hard to read. Also check the legend, I am not sure what the orange circle with black dots represents since it is not mentioned anywhere in the text. If they are not necessary please remove them from the map.*

Corrected according to the reviewer's recommendation.

*Figure 3: Please re-arrange order in the caption so as to present panels in alphabetical order, I believe this would make the reading easier. I also suggest removing the black line since it is not necessary and brings confusion on the plot.*

The captions are corrected according to the reviewer's recommendation. As to the black line, we would prefer keeping it as is. We believe that the black 1:1 line helps the reader to evaluate how far the model is from perfect correspondence with the observed values and in what value range the deviance is detected. In order to further clarify the plot content, we added a line to the figure caption.

*Figure 5: I find the legend confusing since black triangles and black circles supposedly represent ESP and WG based forecasts in general while at the same time represent ESP and WG based forecasts for "number of days with inflow discharge above maximum (NqMax)". Instead of representing "W", "Qmax", "Nq" and "Nqmax" with coloured squares, which are not observed after in the Taylor diagram, I suggest to use a different type of polygon or symbol representing "W", "Qmax", "Nq" and "Nqmax", maybe empty instead, and to colour them on the basis of ESP or WG based forecasts. I also suggest making polygons or symbols larger, since they look rather small compare to the entire graph. In this way, one colour defines whether the forecast is ESP of WG based and the polygon/symbol indicates type of forecast.*

Corrected according to the reviewer's recommendation.

*Figure 7: Please re-arrange order in the caption so as to present panels in alphabetical order, I believe this would make the reading easier. Also describe in the caption what is the black continuous line representing.*

Corrected according to the reviewer's recommendation.

*Figure 9: Colours described in the caption do not match the observed in the figure. Please correct.*

Corrected according to the reviewer's recommendation.

*Figure 10: I am not sure if is a resolution problem or type of file format, however colours are kind of washed out in this figure. It would be better to see it as the other figures.*

Corrected according to the reviewer's recommendation.

*Table 1: I would prefer to see the measurement units on the first column in between parenthesis.*

Corrected according to the reviewer's recommendation.

*Table 2: Same as table 1.*

Corrected according to the reviewer's recommendation.

*References:P28.L25: Replace "Sum-mary" with "Summary"*

Corrected according to the reviewer's recommendation.